# Microglial Autophagy and Mitophagy in Ischemic Stroke: From Dual Roles to Therapeutic Modulation

**DOI:** 10.3390/biology14091269

**Published:** 2025-09-15

**Authors:** Juan Wu, Jiaxin Liu, Yanwen Li, Fang Du, Weijia Li, Karuppiah Thilakavathy, Jonathan Chee Woei Lim, Zhong Sun, Juqing Deng

**Affiliations:** 1Medical School, Kunming University of Science and Technology, Kunming 650500, China; 20232136006@stu.kust.edu.cn (J.W.); 20130141@kust.edu.cn (J.L.); 20232136055@stu.kust.edu.cn (Y.L.); sammuldufang@163.com (F.D.); 2Department of Biomedical Sciences, Faculty of Medicine & Health Sciences, Universiti Putra Malaysia (UPM), Serdang 43400, Selangor, Malaysia; gs72391@student.upm.edu.my (W.L.); thilathy@upm.edu.my (K.T.); 3Genetics and Regenerative Medicine Research Centre, Universiti Putra Malaysia (UPM), Serdang 43400, Selangor, Malaysia; 4Department of Medicine, Faculty of Medicine & Health Sciences, Universiti Putra Malaysia (UPM), Serdang 43400, Selangor, Malaysia; cheewoei@upm.edu.my; 5School of Public Health, Kunming Medical University, Kunming 650500, China; 6Department of Animal Zoology, Kunming Medical University, Kunming 650500, China

**Keywords:** microglia, autophagy, mitophagy, ischemic stroke, neuroinflammation, PINK1/Parkin, inflammasome, M1/M2 polarization

## Abstract

Stroke happens when blood flow to the brain is suddenly blocked, causing damage to brain cells. After a stroke, special immune cells in the brain called microglia become active. These cells can either help repair the brain or make the damage worse. One way microglia respond is by cleaning up damaged parts inside the cell, especially damaged energy centers called mitochondria. This cleaning process, called cellular recycling, can help reduce harmful inflammation and protect brain tissue. However, if the process is too strong or poorly controlled, it may also cause more harm by increasing inflammation and killing healthy cells. In this review, we looked at recent studies to understand how this recycling system works in microglia during and after stroke. We found that keeping the process balanced is key: gentle activation early after stroke can be helpful, while too much later on may be damaging. Some treatments that support this balance, like plant-based compounds or small molecules, show promise in animals but need more testing in humans. Understanding this process may help scientists design better treatments to improve recovery and reduce long-term brain damage in stroke survivors.

## 1. Introduction

Stroke remains one of the leading causes of mortality and long-term disability worldwide, with ischemic stroke (IS) accounting for nearly 85% of all cases. According to the most recent Global Burden of Disease (GBD) Study 2021, the incidence of stroke continues to rise, particularly among younger populations, underscoring its persistent public health burden [1]. Despite significant advances in reperfusion therapies—such as intravenous thrombolysis with recombinant tissue plasminogen activator (rt-PA) and mechanical thrombectomy—these interventions remain constrained by narrow therapeutic time windows, risks of hemorrhagic transformation, and limited accessibility worldwide [2,3]. Thus, adjunctive strategies that extend beyond vascular recanalization and target downstream injury mechanisms are urgently needed.

Beyond neurons, microglia—the resident immune cells of the central nervous system—are increasingly recognized as pivotal regulators of ischemic pathology. They orchestrate neuroinflammation, shape synaptic plasticity, and contribute to angiogenesis, neurogenesis, and functional recovery. Their dual roles, encompassing both pro-inflammatory and reparative phenotypes, have made them attractive therapeutic targets [4,5].

Following ischemic insult, microglia rapidly respond to environmental cues by adopting either a neuroprotective (M2-like) or pro-inflammatory (M1-like) phenotype [6]. This phenotypic polarization is tightly regulated by intracellular stress responses, including autophagy—a conserved catabolic process essential for cellular homeostasis and adaptive immunity [7,8]. Recent studies have revealed that autophagy not only governs microglial activation, metabolism, and phagocytosis but also functions as a molecular switch between protective and detrimental inflammatory responses [7,9,10].

Of particular interest is the role of mitophagy, a specialized form of autophagy that selectively removes damaged mitochondria and has emerged as a metabolic checkpoint for microglial fate under ischemic stress [11,12]. Importantly, the dual nature of autophagy—acting as both a protective and a detrimental process—has been well documented in ischemia–reperfusion injury and microglial responses [13,14,15]. While moderate activation of autophagy can support microglial reprogramming, mitochondrial clearance, and neuroprotection, its dysregulation—whether excessive or insufficient—may exacerbate inflammasome activation, impair phagocytosis, and promote neurotoxicity. This therapeutic paradox is therefore not novel per se, but what remains insufficiently addressed is how these opposing outcomes are determined in a spatiotemporal and cell-type-specific manner after ischemic stroke.

Although several reviews have summarized the general roles of microglial autophagy in ischemic stroke, important gaps remain [15,16,17]. In support of this complexity, have comprehensively reviewed how autophagy exerts divergent effects across brain cell types—including neurons, microglia, astrocytes, and endothelial cells—under ischemic conditions [18]. Their synthesis underscores the need for precision modulation strategies tailored to specific cell populations and injury phases. However, while informative, such reviews often stop short of exploring microglia-specific autophagy in depth or translating these insights into therapeutic design. Prior works have primarily described overarching mechanisms, often without integrating spatiotemporal dynamics, mitophagy-specific pathways, or emerging single-cell and human-derived evidence. Moreover, translational challenges—including blood–brain barrier (BBB) delivery and the lack of reliable biomarkers—are rarely discussed in depth.

At the same time, most experimental studies continue to focus on neuronal or global autophagic responses. By contrast, microglia-specific regulation—encompassing mitochondrial quality control, ER stress, non-coding RNAs, and extracellular vesicles—has not yet been comprehensively examined within a translational framework that links mechanistic insights to clinical feasibility [19,20].

Building on these gaps, the present review offers a focused synthesis that emphasizes the spatiotemporal regulation of microglial autophagy and mitophagy, their integration with immunometabolism and inter-glial communication, and the major translational hurdles to clinical application. We highlight why promising candidates have remained confined to preclinical studies and how next-generation approaches may bridge this divide by combining mechanistic precision with targeted delivery and patient stratification.

Importantly, we draw attention to a recurring phenomenon in this literature: the non-monotonic, U-shaped relationship between autophagy and neuroprotection. In this context, U-shaped denotes a biphasic association in which both insufficient and excessive autophagic activity can be detrimental, whereas moderate activation supports mitochondrial integrity and inflammation resolution. Recognizing this pattern is critical for interpreting the dual roles of microglial autophagy and for guiding therapeutic strategies with phase-specific precision.

To ensure methodological rigor, this study was conducted as a scoping review following established frameworks for evidence mapping. Systematic searches were performed in PubMed and EMBASE up to August 2025, using both controlled vocabulary and free-text terms related to “microglia,” “autophagy,” and “ischemic stroke.” Eligible studies included in vitro experiments, in vivo animal models, and clinical observations investigating mechanistic roles or therapeutic implications. A PRISMA-style flow diagram summarizing the study selection process is shown in Figure 1, and full search strategies are provided in the Appendix A.

In summary, this review highlights the spatiotemporal and context-dependent regulation of microglial autophagy and mitophagy, their integration with immunometabolism, angiogenesis, and neurogenesis, and the translational challenges that have thus far limited clinical application. By focusing on microglia as central regulators of neuroinflammation and tissue repair, we aim to outline novel therapeutic opportunities and unresolved gaps that distinguish this work from prior reviews.

## 2. Dual Roles of Microglial Autophagy in Ischemic Stroke: From Neuroprotection to Neurotoxicity

### 2.1. Protective Role of Microglial Autophagy

Following an ischemic stroke, the abrupt interruption of cerebral blood flow leads to deprivation of oxygen and glucose, resulting in a cascade of energy failure, reactive oxygen species (ROS) accumulation, and cellular organelle dysfunction—particularly affecting mitochondrial integrity and proteostasis [21]. In response to these metabolic insults, microglia—the resident immune cells of the central nervous system (CNS)—are rapidly activated and initiate autophagy as a neuroprotective mechanism [22].

Autophagy, a lysosome-dependent catabolic process, is activated to degrade damaged proteins and organelles, recycle cellular materials, and mitigate oxidative and inflammatory damage [23]. In the early phase of ischemic stroke, microglial autophagy is triggered by metabolic and inflammatory stress. Although its role is context-dependent, growing evidence indicates that autophagy maintains cellular homeostasis and regulates neuroinflammation, conferring neuroprotection under specific conditions [16,17].

Within the ischemic core, severe ATP depletion and calcium overload occur due to the failure of ion transporters such as Na^+^/K^+^-ATPase and Ca^2+^-ATPase [24]. The resulting intracellular Ca^2+^ influx stimulates autophagy via stress-activated pathways, including CaMKKβ-mediated activation of AMP-activated protein kinase (AMPK), which in turn phosphorylates Unc-51-like kinase 1 (ULK1) [25,26,27]. Simultaneously, a reduced ATP/AMP ratio further enhances AMPK activity, which inhibits the mechanistic target of rapamycin (mTOR)—a master negative regulator of autophagy—thereby promoting autophagosome formation [28,29,30].

Emerging evidence suggests that ischemia-driven metabolic reprogramming also shapes autophagic responses in microglia. Under oxygen–glucose deprivation, a metabolic shift occurs from oxidative phosphorylation to glycolysis, accompanied by activation of the AMPK–mTOR–HIF-1α pathway [31,32,33]. This signaling cascade not only facilitates autophagy induction but also promotes M2-like microglial polarization, as shown in several ischemic models [34].

The ischemia-induced unfolded protein response (UPR) also contributes to autophagy activation. Prolonged endoplasmic reticulum (ER) stress, caused by unresolved protein misfolding, activates UPR signaling—particularly IRE1 and PERK—which in turn engage the JNK cascade. Activated JNK phosphorylates Bcl-2, disrupting its interaction with Beclin-1 and thereby initiating Beclin-1–dependent autophagy [35,36]. Furthermore, ischemic injury promotes lysosomal degradation of mTOR, especially in hippocampal neurons, resulting in robust autophagic flux in neurons fated for death [29].

At the cellular level, microglial autophagy exerts protective effects by limiting inflammatory responses and maintaining redox balance. Upon ischemic insult, microglia shift toward an M1-like phenotype, characterized by elevated secretion of pro-inflammatory cytokines such as IL-1β, TNF-α, and inducible nitric oxide synthase (iNOS), all of which worsen neuronal injury [37,38]. Autophagy counters this inflammatory response by reducing cytokine production and promoting a shift toward an M2-like state [39,40].

Additionally, autophagy serves as a key modulator of oxidative stress in ischemic microglia. In oxygen–glucose deprivation/reperfusion (OGD/R) models, microglia produce high levels of ROS, contributing to neuronal apoptosis. Autophagy mitigates ROS toxicity by facilitating mitochondrial clearance and restoring redox homeostasis, thereby attenuating downstream inflammatory signaling [12,41,42].

Although mitophagy plays a central role in this process, detailed mechanisms—such as PINK1/Parkin and BNIP3 signaling—are addressed in Section 2.3. Here, we emphasize the upstream regulators and broader protective outcomes of autophagy activation in microglia.

Microglial autophagy also facilitates the clearance of intracellular protein aggregates via lysosome-dependent mechanisms. For instance, the microglial receptor TREM2 (triggering receptor expressed on myeloid cells 2) interacts with sphingosine-1-phosphate (S1P), enhancing the uptake of misfolded or aggregated proteins [43]. TREM2 signaling is increasingly recognized as a regulator of microglial autophagy–lysosomal function. Its deficiency impairs this system, resulting in toxic substrate accumulation and intensified neuroinflammation [44].

One key transcription factor is hypoxia-inducible factor-1α (HIF-1α), which is stabilized under hypoxic stress. HIF-1α upregulates the expression of BNIP3 (BCL2/adenovirus E1B-19 kDa protein-interacting protein 3), a dual-function molecule. By disrupting the Beclin-1/Bcl-2 complex, BNIP3 frees Beclin-1, allowing autophagosome formation to proceed under ischemic conditions [45,46,47].

Another important regulatory element is the α7 nicotinic acetylcholine receptor (α7nAChR), well known for its anti-inflammatory role in microglia. Activation of α7nAChR favors M2-like polarization and stimulates autophagy, promoting inflammation resolution [48,49]. Specifically, α7nAChR engagement reduces cytokine output through the JAK2/STAT3 signaling pathway [50,51], and concurrently induces COX-2 and prostaglandin E_2_ (PGE_2_) expression, further contributing to anti-inflammatory reprogramming [52].

Phosphodiesterase 1B (PDE1-B) has emerged as a negative regulator of microglial autophagy. Inhibition of PDE1-B by vinpocetine increases autophagic activity in OGD/R-treated BV2 microglia, which in turn elevates the secretion of neuroprotective extracellular vesicles (EVs) marked by CD63 and TSG101 [40]. These EVs are internalized by neurons and diminish OGD-induced neuronal damage, suggesting an autocrine/paracrine axis whereby autophagy enhances microglia–neuron communication (Figure 2).

Beyond microglia, extracellular vesicles from astrocytes, oligodendrocyte precursor cells, endothelial cells, and pericytes—as well as neuron- and stem cell-derived exosomes—have been reported to exert neuroprotective effects in hypoxic–ischemic models by regulating autophagy, inflammation, and neurovascular remodeling [53,54].

Poly(ADP-ribose) polymerase 14 (PARP14) represents another key regulator that simultaneously mediates anti-inflammatory and pro-autophagic functions. In photothrombotic stroke models, PARP14 overexpression suppresses LPAR5 (lysophosphatidic acid receptor 5)-dependent inflammatory signaling, reduces microglial activation, and enhances macroautophagic flux. Conversely, PARP14 knockout intensifies neuroinflammation and disrupts autophagy balance [38].

Recent studies have uncovered competing endogenous RNA (ceRNA) networks that regulate autophagy in ischemic brain cells. For example, in brain microvascular endothelial cells (BMECs), the long non-coding RNA (lncRNA) MALAT1 acts as a miR-26b sponge, thereby derepressing ULK2 and initiating autophagy under OGD/R conditions [55]. Transcriptomic analyses of stroke tissues have identified several lncRNA–miRNA–mRNA axes—such as MALAT1–miR-26b–ULK2—that potentially regulate autophagy and inflammation across diverse cell types, including microglia [56].

Beyond BMECs, lncRNA-mediated ceRNA networks also regulate autophagy and inflammation in microglia. Notably, Tug1 mitigates post-stroke microglial pyroptosis by promoting PINK1/Parkin-dependent mitophagy. Mechanistically, Tug1 stabilizes PINK1 expression and facilitates mitochondrial clearance, thereby suppressing NLRP3 inflammasome activation, caspase-1 cleavage, and IL-1β release. This underscores a functional link between mitophagy and pyroptosis, positioning lncRNA Tug1 as a crucial upstream regulator of microglial fate under ischemic conditions [57].

Another regulatory axis involves the miR-499-5p/PDCD4/ATG5 pathway, which is activated by α-asarone—a naturally occurring compound with known neuroprotective properties. In both OGD/R-treated neurons and middle cerebral artery occlusion (MCAO) models, α-asarone upregulates miR-499-5p, which in turn suppresses PDCD4, a known autophagy inhibitor, while simultaneously upregulating ATG5. This dual action enhances LC3-II accumulation and autophagic flux, reduces neuronal apoptosis, and facilitates functional recovery after stroke [58].

The role of cellular senescence has also gained attention in the context of post-stroke neuroinflammation. Ischemic injury induces a senescence-associated secretory phenotype (SASP), characterized by sustained secretion of inflammatory mediators such as IL-6, IL-1β, TNF-α, and chemokines like CCL3. These pro-inflammatory factors can impair autophagy regulation in microglia [59,60]. Transcriptomic profiling reveals persistent upregulation of SASP-related genes in chronically activated microglia, suggesting a transition to senescence-driven autophagy dysfunction. Targeting SASP components—such as CCL3 or LCP1—is under investigation as a strategy to restore autophagy and resolve chronic inflammation.

In addition, metabolic dysregulation—including disturbances in glucose, lipid, and amino acid pathways—not only alters the inflammatory phenotype of microglia but also directly impairs their autophagic capacity. This interplay between metabolic dysfunction and autophagy impairment has been increasingly associated with poor stroke outcomes. Recent evidence suggests that abnormal microglial autophagy contributes to post-stroke cognitive impairment (PSCI) by disrupting immune–metabolic homeostasis and amplifying neuroinflammation [61].

Together, these findings highlight that microglial autophagy is a highly dynamic and context-sensitive process, shaped by factors such as energy status, redox balance, transcriptional regulation, and cellular aging. These adaptive responses are most pronounced during the acute phase (within 24 h post-stroke), when AMPK activation, HIF-1α stabilization, and early mitophagy cooperate to restore mitochondrial function and promote anti-inflammatory polarization. Therefore, targeting autophagy-regulating pathways—such as AMPK activators, α7nAChR agonists, PDE1-B inhibitors, or non-coding RNA mimics—may provide phase-specific neuroprotection and facilitate resolution of post-ischemic neuroinflammation.

### 2.2. Detrimental Consequences of Dysregulated Microglial Autophagy in Ischemic Stroke

As ischemia progresses into the subacute and chronic phases, persistent or dysregulated autophagy begins to exert maladaptive effects. While microglial autophagy initially plays a protective role, its excessive or prolonged activation—particularly during the reperfusion phase—can paradoxically worsen neuroinflammation and neuronal injury. Interestingly, recent evidence suggests that sustained but moderate autophagic activity may still confer neuroprotection by skewing microglial polarization toward an anti-inflammatory phenotype. Zeng et al. demonstrated that cellular prion protein (PrP^C^) enhances and prolongs autophagy in OGD/R-challenged microglia, which in turn promotes M2-like polarization and reduces neuronal damage [62]. PrP^C^ overexpression elevated LC3B-II/I and LAMP1 levels, while PRNP knockout abolished these changes and reversed the anti-inflammatory phenotype, underscoring the importance of sustained autophagic flux in supporting reparative microglial responses [63]

By contrast, SphK1-driven autophagy in microglia exacerbates neuronal apoptosis and expands infarct volume by activating the TRAF2 pathway following cerebral ischemia–reperfusion, identifying SphK1 as a pro-inflammatory autophagy regulator [62] Moreover, both pharmacological and genetic inhibition of autophagy—such as via 3-MA treatment or Beclin-1 knockout—unexpectedly intensify microglial inflammatory responses and aggravate secondary brain injury [64,65]. Excessive STING-mediated autophagy drives microglial polarization toward a pro-inflammatory phenotype, while STING deficiency restores an anti-inflammatory profile and mitigates injury in ischemia/reperfusion (I/R) models [66].

Recent studies highlight a U-shaped relationship between autophagy and neuroprotection: basal or moderately activated autophagy supports mitochondrial quality control and suppresses inflammation, whereas excessive or persistent autophagic flux can trigger autosis—a Na^+^/K^+^-ATPase-dependent form of cell death characterized by plasma membrane rupture, cytoplasmic vacuolization, and resistance to caspase inhibition [67,68].

The severity of energy depletion and regional metabolic stress sets the threshold at which autophagy transitions from adaptive to maladaptive. In the ischemic core—characterized by profound ATP depletion, calcium overload, and disrupted ion homeostasis—persistent autophagic activation contributes to neuronal death through mechanisms such as lysosomal rupture, membrane disruption, and impaired mitochondrial clearance [9]. In contrast, within the ischemic penumbra where residual perfusion persists, autophagy exerts more nuanced effects. Moderate or transient autophagic flux in this region may help maintain mitochondrial homeostasis and limit oxidative injury, while excessive activation—especially triggered by oxidative bursts during reperfusion—can exacerbate inflammation or induce programmed cell death [69,70].

The regional heterogeneity of the ischemic brain further modulates the dual role of autophagy. In the penumbra—a dynamic zone with partial perfusion—autophagy may initially provide protection against apoptosis and help preserve mitochondrial function. However, prolonged or excessive autophagic activation in this region—particularly following delayed reperfusion or recurrent ischemia—can shift the balance toward autophagy-dependent neuronal death. This maladaptive transition contributes to the expansion of infarcted tissue, a phenomenon often associated with “futile reperfusion” in stroke therapy [69,71].

A particularly destructive mechanism arises when dysregulated autophagy fails to restrain inflammasome activation. The accumulation of autophagic substrates such as p62/SQSTM1 enhances NF-κB signaling, leading to elevated production of IL-1β and IL-18 via caspase-1 activation. This inflammatory cascade is further amplified by mitochondrial DNA (mtDNA) leakage, which activates the cGAS–STING axis—a cytosolic DNA-sensing pathway (Figure 3). STING facilitates LC3 lipidation at ERGIC-derived membranes and promotes noncanonical autophagy in an ATG5-dependent manner, thereby enhancing LC3-II conversion and sustaining downstream cytokine release and inflammasome activation [72,73,74].

Critically, STING directly interacts with LC3 to establish a molecular bridge between autophagy and inflammation in microglia. In ischemic stroke models, overactivation of STING enhances LC3 lipidation at endoplasmic reticulum–Golgi intermediate compartments (ERGICs), thereby intensifying autophagic flux and driving microglial polarization toward the pro-inflammatory M1 phenotype. This response amplifies inflammasome activation and increases infarct volume. Conversely, STING knockout or pharmacological inhibition restores LC3 homeostasis, promotes M2 polarization, reduces inflammatory cytokine release, and improves neurological outcomes after stroke [66,74].

Moreover, TLR9—an endosome-localized DNA sensor—detects mitochondrial DNA fragments released during oxidative stress, thereby complementing the cGAS–STING axis in sustaining inflammation. Activation of TLR9 by circulating mtDNA amplifies pro-inflammatory signaling through the MyD88–NF-κB pathway and facilitates inflammasome assembly and production of IL-1β and IL-18. These autophagy–inflammation feedback loops accelerate damage propagation into the ischemic penumbra, where the inflammatory environment compromises neurons that might otherwise be salvageable [75].

Another major contributor to autophagy dysregulation is prolonged endoplasmic reticulum (ER) stress [76]. The PERK–eIF2α–CHOP signaling axis, typically activated during ischemia–reperfusion, suppresses global protein synthesis and promotes autophagy-mediated cell death [76,77]. Chronic activation of this pathway shifts autophagy from a protective to a maladaptive process, fostering microglial inflammation instead of resolution [77]. In experimental models of cerebral ischemia, inhibition of PERK signaling reduces M1-polarized microglial activation and neuroinflammation, highlighting its pathogenic role in maladaptive autophagic responses [20].

The enzyme protein tyrosine phosphatase 1B (PTP1B) links ER stress to sustained microglial activation. Overexpression of PTP1B enhances inflammatory cytokine production and impairs the M1-to-M2 phenotypic switch, thereby hindering post-ischemic recovery. Conversely, pharmacological inhibition of PTP1B alleviates ischemic neuronal injury by modulating the PERK-mediated ER stress–autophagy axis in microglia, reducing excessive autophagy and dampening neuroinflammation [20]. Likewise, TMEM166, a transmembrane autophagy regulator, suppresses the Akt pathway, thereby aggravating inflammatory damage via IL-6 and CRP upregulation—both of which are predictive of poor stroke prognosis [78,79].

In parallel, intercellular communication between microglia and neurons becomes disrupted during autophagy dysregulation. Neurons experiencing excessive autophagy reduce the secretion of CX3CL1—a key “off-signal” that normally restrains microglial activation [80]. At the same time, autophagy-activated microglia downregulate CX3CR1, the corresponding receptor for CX3CL1, thereby disrupting this crucial signaling axis and promoting sustained inflammation [81]. In the ischemic penumbra, the loss of homeostatic neuron–microglia signaling exacerbates neuronal vulnerability and encourages maladaptive microglial phenotypes. Furthermore, miR-30d, an ischemia-induced microRNA, upregulates Beclin-1, triggering autophagy-dependent neuronal death [82]. The resulting neuronal debris intensifies microglial activation, completing a self-reinforcing pathological loop.

An additional paradox of dysregulated autophagy lies in phagocytic misdirection. Under normal conditions, microglia selectively clear apoptotic or damaged cells. However, during neuroinflammatory stress, upregulated autophagy may promote phagoptosis—a process in which viable neurons and intact myelin are aberrantly engulfed. This maladaptive response is facilitated by disrupted coordination between autophagy-related proteins, such as Beclin-1 and LC3, and phagocytic receptors like TREM2. Notably, Beclin-1 is required for the recycling of TREM2 and other receptors to the microglial surface; its deficiency leads to impaired recognition of “eat-me” signals and contributes to non-selective phagocytosis of living neural components [83]. Such dysfunctional clearance mechanisms aggravate neurodegeneration, especially in the ischemic penumbra, where persistent inflammation impedes resolution [84].

Markers such as Galectin-3 and complement C1q further tag functionally salvageable neurons for removal, accelerating inappropriate synaptic pruning and contributing to long-term cognitive impairment [85,86].

Reactive oxygen species (ROS) accumulation and defective mitophagy compound the problem, forming a self-sustaining ROS–autophagy–inflammasome feedback loop. Elevated ROS levels activate autophagy via AMPK and MAPK pathways; however, when autophagic flux is impaired, ROS production remains unchecked due to continued mitochondrial damage. This leads to mitochondrial DAMP release, triggering NLRP3 inflammasome activation, which in turn drives further ROS production, perpetuating a pathological cycle [41,87].

This pathological cycle sustains activation of NLRP3 inflammasomes, enhances NF-κB transcription, and drives persistent cytokine release, thereby contributing to chronic neuroinflammation [88,89]. Meanwhile, lysosomal stress caused by continuous autophagic burden further compromises intracellular degradation systems, tipping the balance toward irreversible neurodegeneration [90,91].

These maladaptive processes exacerbate not only gray matter injury but also white matter damage by disrupting oligodendrocyte survival and compromising myelin integrity. For instance, microglial autophagy—particularly via the TLR4–STAT1/6 signaling pathway—aggravates ischemic white matter injury and impairs oligodendrocyte function [92]. Moreover, chronic microglial activation hinders long-term myelination and oligodendrocyte maturation, whereas microglial depletion facilitates remyelination and functional recovery after stroke [5]. Additionally, intrinsic autophagy within oligodendrocyte lineage cells is essential for maintaining myelin integrity and glial cell survival, establishing a mechanistic link to sensorimotor and cognitive deficits in aging and post-ischemic conditions [93].

Given this complex landscape, microglial autophagy should be understood as a context-sensitive mechanism, whose effects are shaped by regional ischemic severity, temporal dynamics, and cellular environmental cues. Therefore, simplistic strategies that uniformly promote or inhibit autophagy—without accounting for these contextual variables—are unlikely to yield consistent therapeutic benefits [16,17].

Instead, emerging therapeutic strategies aim to recalibrate autophagic flux with temporal and spatial precision. For example, STING inhibitors [66], PTP1B blockers [20], and miRNA-based modulators such as anti-miR-30d [94] are under investigation for their ability to restore autophagic balance and reduce neuroinflammation. Additionally, the development of dynamic biomarkers—such as autophagosome-to-lysosome ratios or autosis markers—may enable real-time monitoring of autophagic activity, paving the way for personalized interventions.

Therapeutic interventions targeting autophagy must be tailored to the specific phase of stroke and regional injury dynamics. In the early phase (0–24 h), induction of autophagy—via upstream regulators such as AMPK or mTOR inhibition—has demonstrated neuroprotective effects, primarily by mitigating mitochondrial damage and suppressing inflammasome priming [95,96]. In particular, PINK1/Parkin-mediated mitophagy enhances mitochondrial clearance and supports inflammation resolution under ischemic stress. By contrast, during the subacute and chronic stages, sustained autophagy—especially driven by STING or ER stress–CHOP signaling—has been associated with autosis, phagoptosis, and persistent neuroinflammation [66,97]. This has led to the proposal of a “biphasic” therapeutic strategy, wherein early-phase activation is followed by later-phase modulation or suppression, potentially yielding optimal outcomes. Support for this concept comes from studies involving both pharmacologic agents (e.g., STING inhibitors, rapamycin) and molecular modulators [98,99].

In sum, dysregulated autophagy serves as a central integrative mechanism in the propagation of ischemic injury, interacting with mitochondrial dysfunction, immune amplification, ER stress, apoptosis, and intercellular miscommunication. Effective therapeutic targeting must therefore aim to restore homeostatic autophagy without triggering maladaptive overactivation. Achieving this requires precisely timed and cell-specific interventions. For instance, early-phase activation of autophagy through AMPK stimulation or transient mTOR inhibition may promote the clearance of damaged mitochondria, whereas late-phase suppression using agents such as 3-MA or chloroquine could help limit excessive catabolism and inflammation [100]. Furthermore, natural compounds like Urolithin A have shown promise in enhancing autophagy, reducing ER stress–related neuronal injury, and inducing mitophagy in non-CNS contexts, thereby preserving mitochondrial integrity [101]. These multilevel, context-driven interventions represent promising avenues for balancing the dual roles of microglial autophagy—both protective and detrimental—as summarized in Table 1.

### 2.3. Microglial Mitophagy in Ischemic Stroke: Mechanistic Crossroads and Therapeutic Promise

As a specialized extension of the protective autophagy mechanisms discussed in Section 2.1, microglial mitophagy refers to a mitochondria-targeted process that plays unique roles in the resolution of ischemic injury. Following cerebral ischemia, microglia undergo profound shifts in metabolism, phenotype, and inflammation, many of which are tightly coupled to mitochondrial homeostasis. As a subtype of autophagy, mitophagy not only clears dysfunctional mitochondria but also regulates immune responses, maintains redox balance, and influences microglial phenotype polarization. Given that ischemia-induced mitochondrial dysfunction simultaneously contributes to neurodegeneration and neuroinflammation, mitophagy emerges as a central mechanism linking immunometabolic control and injury resolution [11,109,110].

Mitochondria-specific autophagy (mitophagy), primarily mediated through the PINK1/Parkin and BNIP3 pathways, is essential for maintaining mitochondrial quality control during ischemic injury. By selectively removing dysfunctional mitochondria, mitophagy suppresses NLRP3 inflammasome activation and attenuates pro-inflammatory signaling [12,42]. This targeted clearance not only reduces reactive oxygen species (ROS) accumulation but also prevents the release of mitochondrial DNA (mtDNA) and other damage-associated molecular patterns (DAMPs) that would otherwise amplify neuroinflammation. Concurrently, ischemia-induced zinc accumulation disrupts mitochondrial integrity, promotes excessive ROS production, and triggers maladaptive autophagy via mTOR inhibition and oxidative stress enhancement, thereby intensifying neuroinflammatory cascades [111].

The canonical PINK1/Parkin signaling axis is among the most extensively characterized pathways regulating mitophagy in microglia. Under conditions of mitochondrial stress, PINK1 accumulates on the outer mitochondrial membrane and recruits Parkin, which mediates ubiquitination of damaged mitochondrial proteins. This modification facilitates the recruitment of autophagy receptors, including p62 and NDP52, which in turn promote the formation of autophagosomes and subsequent mitochondrial degradation [112,113].

Autophagy attenuates ROS-induced cytotoxicity by selectively degrading dysfunctional, ROS-generating mitochondria through mitophagy, thereby restoring intracellular redox homeostasis and suppressing subsequent inflammatory responses [12,41,42].

In parallel, BNIP3 and NIX (also known as BNIP3L) function as mitochondrial “eat-me” signals that mediate receptor-dependent mitophagy independently of ubiquitination. These mitophagy receptors interact directly with LC3 through their LC3-interacting region (LIR) motifs and are markedly upregulated under hypoxic conditions or HIF-1α signaling, particularly within the ischemic penumbra.

Multiple upstream molecular regulators have been identified that finely modulate mitophagy activity in ischemic microglia. Among them, hypoxia-inducible factor-1α (HIF-1α) plays a pivotal role under conditions of oxygen deprivation. Stabilized during hypoxic stress, HIF-1α transcriptionally upregulates BNIP3 (BCL2/adenovirus E1B 19-kDa protein-interacting protein 3), a multifunctional protein involved in both macroautophagy and mitophagy. BNIP3 disrupts the inhibitory Beclin-1/Bcl-2 complex, thereby releasing Beclin-1 to initiate autophagosome biogenesis and promote the clearance of depolarized mitochondria [46,47].

Recent studies have further demonstrated that transforming growth factor beta 1 (TGFB1) signaling can promote mitophagy through PLSCR3-mediated cardiolipin externalization, acting in coordination with BNIP3L/NIX, BNIP3, and FUNDC1 receptor-dependent pathways [114]. Supporting this mechanism, clinical cerebrospinal fluid (CSF) samples from patients with subarachnoid hemorrhage have shown upregulation of BNIP3L/NIX during ischemic stress [115].

Mitophagy plays a central role in directing microglial polarization toward either a neurotoxic M1 phenotype or a neuroprotective M2 phenotype. In ischemic models, enhanced mitophagy promotes the clearance of ROS-producing mitochondria, thereby suppressing NLRP3 inflammasome activation and favoring M2-like anti-inflammatory polarization. Microglia-specific overexpression of PGC-1α, a transcriptional coactivator that stimulates ULK1-dependent mitophagy, reduces inflammatory cytokine production and improves neurological outcomes [116] Pharmacological agents such as Sodium Tanshinone IIA Sulfonate (STS) have been shown to modulate autophagic and inflammatory responses in neuron–microglia co-culture systems, contributing to neuroprotection in ischemic models [117]. Mitophagy is initiated early during the acute phase of stroke and continues into the subacute remodeling period, dynamically shaping both redox homeostasis and microglial phenotype over time [118]. Notably, STS modulates mitophagy-associated pathways not only in vitro but also in vivo, where it downregulates LC3-II, Beclin-1, and Sirt6 expression, while simultaneously attenuating inflammatory cell infiltration and neuronal injury in MCAO/R models. These findings underscore the dual regulatory capacity of STS in controlling both autophagy and inflammation, highlighting its potential as a therapeutic agent for ischemic stroke (Figure 4).

In addition to mitophagy, emerging evidence implicates ferroptosis—a regulated, iron-dependent form of cell death marked by lipid peroxidation—as a significant contributor to microglial injury following ischemia/reperfusion. While mitophagy counteracts oxidative stress by eliminating dysfunctional mitochondria, ferroptosis is primarily driven by intracellular glutathione depletion and inactivation of glutathione peroxidase 4 (GPX4), culminating in uncontrolled oxidative damage. N6022, a selective inhibitor of S-nitrosoglutathione reductase (GSNOR), has been shown to mitigate microglial ferroptosis in both MCAO/R rodent models and OGD/R cellular models. It achieves this through dual mechanisms: enhancing Nrf2 nuclear translocation and suppressing the GSNOR–GSTP1 axis. This concerted action boosts antioxidant defenses while preserving glutathione-S-transferase P1 (GSTP1) activity, effectively reducing lipid ROS accumulation and downstream neuroinflammation. Collectively, these findings support the therapeutic potential of targeting the Nrf2–GSNOR–GSTP1 pathway as a complementary approach to mitophagy-based interventions, with the goal of reestablishing redox homeostasis and enhancing microglial neuroprotection in ischemic stroke [119].

A wide array of endogenous molecular regulators orchestrates both the initiation and resolution phases of mitophagy in ischemic microglia, thereby shaping their inflammatory status and determining cellular outcomes. These regulatory molecules—along with their specific mechanistic functions and the stroke stages in which they are predominantly active—are summarized in Table 2.

However, not all evidence supports a uniformly protective role for autophagy. In certain ischemic post-conditioning (IPOC) models, autophagy inhibition with 3-MA reduced infarct size and cerebral edema, whereas induction with Rapamycin abolished these benefits. This supports the notion that autophagy functions as a “double-edged sword,” where timing, magnitude, and cell type critically influence outcomes [15]. Similarly, in microglia, excessive autophagy has been linked to pro-inflammatory polarization, whereas its suppression may attenuate injury in specific contexts. For instance, permanent MCAO-induced hypoxia has been shown to elevate microglial autophagy at lesion borders, thereby exacerbating neuroinflammation and neuronal damage. In these models, pharmacological inhibition with 3-MA significantly decreased microglial activation, infarct volume, cerebral edema, and neurological deficits, highlighting the pathological potential of overactivated autophagy under defined ischemic conditions [128].

Conversely, inhibition or exhaustion of mitophagy results in the accumulation of dysfunctional mitochondria, persistent ROS production, and heightened NF-κB-driven transcription of pro-inflammatory mediators such as IL-1β and TNF-α. Notably, excessive mitochondrial DNA leakage under these conditions activates the cGAS-STING axis, sustaining inflammatory responses and disrupting microglia-neuron cross-talk [72,129]. In parallel, HMGB1—another archetypal DAMP actively released by microglia and passively from necrotic cells—further amplifies neuroinflammation by engaging TLR4 and RAGE receptors. These downstream cascades converge on NLRP3 inflammasome activation, MMP9 upregulation, and blood–brain barrier disruption, thereby intensifying cerebral edema and elevating hemorrhagic risk. Interestingly, HMGB1 exerts a biphasic role in stroke: while driving acute-phase inflammation, it also facilitates late-stage neurovascular remodeling and repair, underscoring its temporal complexity and therapeutic relevance [130].

Recent studies highlight the pivotal roles of non-coding RNAs in orchestrating mitophagy-related signaling cascades. For instance, lncRNA MALAT1 acts as a competing endogenous RNA (ceRNA) by sponging miR-26b, thereby derepressing ULK2, a key initiator of mitophagy, and enhancing autophagic flux in ischemic models [55]. Similarly, MALAT1 can also sponge miR-30a, releasing translational inhibition on Beclin-1, which promotes autophagosome formation and reduces neuronal apoptosis following stroke [122]. Together, these findings position MALAT1 as a dual regulator of mitophagy, simultaneously acting through ULK2 and Beclin-1 pathways.

Exosome-derived miRNAs such as miR-126 and miR-223 play essential roles in regulating mitophagy flux and microglial phenotype. miR-126 exosomes, derived from adipose-derived stem cells (ADSCs), target components of the PI3K/Akt/mTOR axis, promoting autophagy and suppressing NF-κB-mediated inflammation in ischemic models [123]. Meanwhile, miR-223-3p exosomes inhibit microglial M1 polarization by repressing STAT3 signaling, thus indirectly promoting mitophagy and resolving inflammation [124]. These exosomal miRNAs exemplify how intercellular communication channels can fine-tune microglial mitophagy and immune responses.

In addition to non-coding RNAs, pharmacological agents such as geniposide also modulate microglial polarization and autophagic activity. Geniposide has been shown to target the SOX2/RIPK1 signaling axis, thereby inhibiting M1 polarization and inflammatory cytokine release in OGD/R-exposed BV2 microglia and in MCAO mouse models. This action restores autophagic homeostasis and enhances neuronal protection, establishing SOX2–RIPK1 as a novel mechanistic link between inflammation and autophagy in ischemic stroke [131].

Beyond pharmacological interventions, non-pharmacological strategies such as exercise rehabilitation show potential in modulating microglial autophagy. Recent studies in animal and in vitro models demonstrate that aerobic exercise increases exosome release—notably enriched in neuroprotective miRNAs such as miR-124—from microglia and neurons. These exosomes suppress pro-inflammatory signaling while enhancing mitophagy flux and antioxidative responses in peri-infarct regions. For example, exercise preconditioning in rats elevates circulating exosome miR-124 levels and attenuates neuronal apoptosis via STAT3/BCL-2/BAX pathways [132]. Likewise, microglial exosome–derived miR-124-3p, validated in traumatic brain injury models, suppresses STAT3-dependent mTOR activation, thereby enhancing mitophagic clearance of damaged mitochondria and promoting M2-like microglial polarization in ischemia [133]. Although human data remain limited, the ADEX randomized controlled trial in Alzheimer’s disease patients showed that 16 weeks of aerobic exercise modulated neuron-derived extracellular vesicle cargo (e.g., BDNF), supporting translational relevance [134]. Taken together, these findings underscore the therapeutic relevance of exercise-induced exosomal miRNA delivery systems targeting mitophagy-regulatory nodes, while highlighting the need for further human validation.

DRP1, a fission mediator, is rapidly activated during early ischemia, promoting the fragmentation of damaged mitochondria, which may aid their removal via mitophagy. Interestingly, even when DRP1 is inhibited, mitophagic flux persists, indicating that mitophagy can occur independently of DRP1-driven fission [125]. While excessive DRP1 activity—driven by ROS and Ca^2+^ overload—clearly exacerbates mitochondrial fragmentation, the assumption that it saturates mitophagic capacity requires further validation. Recent evidence has delineated an upstream regulatory cascade linking transcriptional and kinase control to DRP1-mediated mitochondrial fission in microglia. Zhao et al. identified an E2F1/CDK5/DRP1 signaling axis that orchestrates mitochondrial fragmentation and autophagic activation following cerebral ischemia–reperfusion. Mechanistically, ischemic stress enhances E2F1 activity, which transcriptionally upregulates CDK5 [126]. The latter phosphorylates DRP1, promoting mitochondrial division and facilitating mitophagic clearance. Although initially beneficial for mitochondrial quality control, sustained activation of this cascade aggravates neuroinflammation and microglial overactivation, worsening neuronal injury. Pharmacological inhibition of E2F1, CDK5, or DRP1 alleviated inflammation, reduced infarct volume, and improved behavioral recovery, suggesting this axis as a promising therapeutic target for maladaptive mitophagy in stroke.

Fusion proteins such as MFN2 and OPA1 are essential for mitochondrial network integrity. Their downregulation under ischemic stress contributes to mitochondrial swelling and cristae disruption, impairing respiratory efficiency. Loss of fusion favors glycolysis and promotes pro-inflammatory microglial phenotypes [127].

Therapeutic restoration of mitochondrial quality control is a promising strategy to resolve neuroinflammation after stroke. Natural compounds like ligustilide activate PINK1/Parkin- and BNIP3–LC3-dependent mitophagy, while also engaging PI3K/Akt signaling to reduce oxidative stress and apoptosis, thereby protecting against ischemic neuronal damage [135,136,137]. Conversely, excessive mitophagy via BNIP3–LC3 may exacerbate delayed neuronal death [138]. Additionally, mitochondrial-targeted molecules such as SS-31 (Elamipretide)—which binds cardiolipin, restores autophagic flux, and reduces oxidative damage in traumatic brain injury and non-CNS ischemia models—and Urolithin A—which activates autophagy and suppresses ER stress in ischemic models, while inducing mitophagy in non-CNS contexts such as muscle and Parkinson’s disease—exhibit potential for translational application in stroke-related therapies [139,140]. Importantly, urolithin A has demonstrated oral bioavailability and safety in elderly human participants, bolstering its translational appeal [141]. Despite its strong preclinical promise, SS-31 has encountered translational hurdles. In a phase 3 clinical trial involving patients with primary mitochondrial myopathy, elamipretide failed to improve the six-minute walk test or fatigue scores compared with placebo, despite good tolerability [142]. In addition, its application in rare disease contexts—such as Barth syndrome—has faced regulatory setbacks, including FDA rejection of the marketing application in May 2025 due to inadequate clinical evidence, underscoring the challenges of demonstrating efficacy in heterogeneous patient populations [143].

Nonetheless, major hurdles remain. These include the cell-type specificity of mitophagic flux, the optimal timing of intervention, and risks of autophagic overactivation leading to autosis. For instance, the canonical PINK1–Parkin pathway is activated during the acute phase of ischemia to initiate mitophagy, but its regulation differs across cell types and timepoints, leading to divergent outcomes [11]. In the late phase, regulatory networks such as the MALAT1–miR-26b–ULK2 axis can reshape mitophagy activity based on cellular context and recovery needs, emphasizing phase-specific plasticity [144]. Future therapeutic strategies must therefore be finely tuned to preserve beneficial mitophagic flux while preventing detrimental overactivation.

Microglial mitophagy is a critical determinant of ischemic outcome, functioning as a metabolic rheostat and inflammatory checkpoint. It bridges mitochondrial dynamics, redox balance, phenotypic polarization, and neuroimmune interactions. Targeting this pathway holds translational promise for stroke therapies, especially when combined with epigenetic and exosome-based delivery platforms.

While preclinical studies have established a compelling role for microglial autophagy and mitophagy in ischemic stroke, translational challenges remain. Most available evidence derives from rodent models or immortalized microglial lines, which may not faithfully capture the transcriptomic, epigenetic, and functional heterogeneity of human microglia. Direct access to patient-derived microglia—especially during the acute and subacute phases of stroke—is limited by ethical and technical constraints. Although emerging single-cell transcriptomic atlases of human post-stroke brain tissue have begun to uncover cell-specific responses, dedicated analyses of microglial autophagy are still lacking [145]. Preliminary clinical observations—such as elevated circulating ATG5 levels in ischemic stroke patients, which correlate with worse functional outcomes—suggest indirect translational relevance but fall short of confirming functional autophagic activity within human microglia [146]. Bridging this gap will require integrated platforms such as iPSC-derived microglia, single-cell/spatial transcriptomics, and advanced imaging to validate mechanistic targets in human contexts.

To date, no therapy selectively modulating microglial mitophagy has been approved or trialed in stroke patients. Representative compounds—including β-elemene, ligustilide, Urolithin A, and SS-31—remain confined to the preclinical stage, highlighting the mechanism–delivery–population gap that continues to impede translation. Moving forward, integration of high-resolution in vivo imaging, patient-derived microglia, and microglia-specific biomarkers will be essential to link mechanistic insights with clinical feasibility. These representative agents and their mechanistic targets are summarized in Table 3, providing a translational snapshot of ongoing preclinical efforts.

Beyond their role in neuroinflammation, microglia are increasingly recognized as active regulators of post-stroke angiogenesis and neurogenesis. They contribute to tissue repair by releasing VEGF, remodeling the extracellular matrix, and promoting synaptic plasticity. However, whether microglial autophagy or mitophagy directly drives these regenerative processes remains unclear. Current evidence suggests that enhancing mitophagy may suppress inflammation and improve phagocytic clearance, thereby indirectly fostering a permissive environment for vascular and neuronal regeneration [5,156]. This gap underscores the need for future studies using microglia-specific autophagy manipulations—such as Atg7 or PINK1 knockouts—combined with angiogenic and neurogenic assays to determine whether mitophagy directly regulates post-ischemic regeneration.

## 3. Discussion

### 3.1. Mechanistic Duality and Temporal Dynamics

This scoping review mapped the current evidence on microglial autophagy and mitophagy in ischemic stroke, integrating mechanistic, pharmacological, and translational findings. Our synthesis highlights both protective and detrimental roles of autophagy, depending on temporal and spatial contexts, and emphasizes its dual potential as a therapeutic target.

Microglial autophagy exhibits a striking duality in ischemic stroke, functioning as both a cytoprotective mechanism and a driver of secondary injury. During the acute ischemic phase, moderate activation facilitates cellular adaptation by clearing damaged mitochondria, reducing oxidative stress, and promoting anti-inflammatory polarization. This adaptive response is mediated in part by AMPK/ULK1 and PI3K/mTOR pathways, which regulate autophagy initiation and resolution. Pharmacological agents such as pseudoginsenoside-F11 (PF11) and myricetin oligomers exemplify this effect by enhancing autophagic flux and shifting microglia toward an M2 phenotype [157,158].

However, when autophagy becomes excessive, prolonged, or dysregulated, it transitions from protective to detrimental. Under such conditions, autophagy facilitates outcomes such as autosis, inflammasome hyperactivation, and neuronal death [9]. The dual role of autophagy is therefore tightly linked to its spatiotemporal regulation. In the acute phase, autophagy is predominantly neuroprotective, mitigating mitochondrial damage and inflammatory amplification in the penumbra. By contrast, in subacute and chronic phases, persistent autophagic stress, impaired flux, and upregulation of SASP factors and ceRNA networks (e.g., MALAT1–miR-26b, CCL3) contribute to sustained neuroinflammation and cognitive decline [55,159]. Spatially, the ischemic core often exhibits autosis and energetic collapse, whereas the penumbra remains responsive to autophagic modulation—highlighting the penumbra as a more promising therapeutic target.

Evidence from both clinical and preclinical models suggests that unresolved autophagic flux exacerbates neuroinflammation through persistent ROS generation, mitochondrial DNA leakage, and maladaptive NLRP3 inflammasome activation [87,160]. This mechanistic duality underscores the need for carefully timed, context-specific interventions that preserve beneficial autophagy while avoiding its pathological overactivation.

### 3.2. Mitophagy as an Immunometabolic Checkpoint

Among the various forms of autophagy, mitophagy—the selective removal of damaged mitochondria—emerges as a pivotal determinant of microglial fate in ischemic stroke. Dysfunctional mitochondria are not only metabolic liabilities but also potent inducers of inflammation through the release of reactive oxygen species (ROS) and mitochondrial DNA (mtDNA). These danger signals activate innate immune sensors such as TLR9, STING, and the NLRP3 inflammasome. Notably, mtDNA leakage can trigger cGAS–STING signaling, which in turn promotes NLRP3 inflammasome assembly and cytokine release, forming a self-amplifying inflammatory loop within ischemic lesions. Efficient mitophagy therefore functions as a cellular filter, maintaining redox balance and suppressing neuroinflammation [72,75,129].

The canonical PINK1/Parkin pathway is the best-characterized mitophagy axis. Upon mitochondrial depolarization, PINK1 accumulates on the outer mitochondrial membrane and recruits Parkin, which ubiquitinates mitochondrial proteins and facilitates their recognition by autophagy receptors [161]. In parallel, receptor-mediated pathways—BNIP3/NIX and FUNDC1—directly engage LC3 via LIR motifs, providing alternative mechanisms independent of ubiquitination [120,121]. These parallel axes ensure that mitophagy remains functional even when Parkin activity is compromised, a scenario frequently observed in ischemic stroke.

Mitophagy also integrates tightly with microglial immunometabolism. Enhanced mitophagy preserves oxidative phosphorylation, sustains ATP homeostasis, and promotes M2 anti-inflammatory polarization. Conversely, impaired mitophagy drives a metabolic switch toward glycolysis, reinforcing pro-inflammatory M1 phenotypes. Moreover, mitophagy interacts with mitochondrial biogenesis programs governed by PGC-1α and SIRT1, maintaining a dynamic balance essential for cellular adaptation under ischemic stress [116].

Taken together, mitophagy acts as an immunometabolic checkpoint, balancing mitochondrial quality control with inflammatory responses. Its central role underscores both the therapeutic promise of targeting microglial mitophagy and the necessity of tailoring interventions to cellular context and temporal dynamics.

### 3.3. Translational Gaps and Drug Landscape

The regulation of microglial autophagy in ischemic stroke is orchestrated by a complex network of signaling pathways with strong temporal and spatial specificity. At the core of this network lies the PI3K/Akt/mTOR axis, which suppresses autophagy under homeostatic conditions. Numerous pharmacological agents—including vinpocetine and β-elemene—induce autophagy by inhibiting mTOR signaling [10,40]. Additionally, flavonoids such as baicalein have demonstrated neuroprotective effects in ischemic stroke models by activating the PI3K/Akt/mTOR pathway, thereby inhibiting excessive autophagy and reducing neuronal apoptosis [97]. In parallel, baicalin, another flavonoid, has been shown to exert anti-inflammatory and metabolic regulatory effects in ischemic injury [162]. Other flavonoids, including quercetin, also downregulate pro-inflammatory cytokines and improve neurological recovery following ischemia, suggesting shared neuroprotective mechanisms with anti-inflammatory overlap [163]. Recent findings further highlight that esketamine, at a clinical dose, can inhibit the Akt signaling pathway, promote microglial M2 polarization, and enhance autophagy, ultimately reducing cerebral ischemia/reperfusion injury [164]. However, broad suppression of mTOR can disrupt lysosomal function, synaptic remodeling, and phagocytosis, underscoring the need for selective rather than indiscriminate modulation.

Other regulators link autophagy with hypoxia, DNA sensing, and endoplasmic reticulum (ER) stress. For instance, HIF-1α upregulates BNIP3 under hypoxic conditions, promoting mitophagy [46]. STING, while initiating autophagy through LC3 recruitment, paradoxically enhances NLRP3 inflammasome activation and pro-inflammatory cytokine release [66]. Similarly, the PERK–eIF2α–CHOP axis connects ER stress to autophagic cell death, and its modulation—such as by PTP1B inhibition or ALKBH5 overexpression—has been shown to attenuate ischemic inflammation [20,79]. These findings emphasize that therapeutic interventions must carefully balance adaptive autophagy versus pathological overactivation. Notably, NOD-like receptor X1 (NLRX1) has emerged as a key autophagy-promoting factor that alleviates cerebral ischemia/reperfusion-induced neuronal injury. Mechanistically, NLRX1 directly binds to ATG5, enhancing autophagic activity while suppressing NLRP3 inflammasome signaling and pro-inflammatory cytokine production. This dual action—supporting protective autophagy while dampening inflammasome-mediated inflammation—identifies the NLRX1–ATG5 axis as a promising therapeutic target in ischemic stroke [165].

In this context, upstream regulators such as ULK1—the initiator kinase of autophagy—have drawn attention for their dual role in neuroprotection and immunomodulation. Xiong et al. demonstrated that pharmacological activation of ULK1 using LYN-1604 significantly reduced infarct volume, improved behavioral outcomes, and promoted anti-inflammatory microglial/macrophage polarization in a photothrombotic stroke model [166]. Conversely, ULK1 inhibition exacerbated neuroinflammation and tissue damage, highlighting its therapeutic potential. In a complementary study, Xiong et al. further showed that genetic deletion of ULK1 exacerbates ischemia-induced microglial dysfunction, marked by impaired myelin debris clearance and heightened pro-inflammatory activity [167]. Mechanistically, ULK1 was found to interact with TRAF6 and modulate NF-κB signaling, establishing it as a key endogenous brake on post-ischemic neuroinflammation.

Epigenetic regulators and non-coding RNAs further shape this landscape. Long non-coding RNAs such as MALAT1 and TUG1, and microRNAs including miR-138-5p, miR-26b, and miR-30d, fine-tune the expression of autophagy-related genes (e.g., PINK1, SIRT1, Beclin-1). Exosomal miRNAs—particularly miR-30d-5p and miR-138-5p—have shown potential in restoring protective autophagy and promoting M2 polarization in preclinical stroke models [55,97,168]. Collectively, these RNA-based mechanisms provide a dynamic and phase-sensitive regulatory layer of microglial plasticity.

Several small-molecule compounds—including salidroside, ursolic acid, and their derivatives—have been reported to modulate autophagy- and mitophagy-related pathways, reduce ROS burden, and promote anti-inflammatory polarization. Salidroside exerts neuroprotective effects in ischemia/reperfusion models by activating the AMPK/TSC2/mTOR pathway, thereby inducing autophagy, reducing oxidative stress, and attenuating neuronal apoptosis [97]. Similarly, ursolic acid modulates microglial polarization indirectly via the PPARγ–MMP2 pathway, restores MMP/TIMP balance, and reduces neuronal injury; in addition, it activates Nrf2/HO-1 signaling to mitigate oxidative stress [148,149,150]. Polyphyllin I (PPI), a steroidal saponin derived from Paris polyphylla, has also been shown to alleviate cerebral ischemia/reperfusion injury by promoting autophagy-mediated M2 polarization of microglia. Mechanistically, PPI suppresses the Akt/mTOR pathway, facilitates ROS clearance, and inhibits NLRP3 inflammasome activation. Pharmacological inhibition of autophagy or reactivation of NLRP3 abolishes its protective effects, underscoring the pivotal role of autophagy in shaping neuroinflammatory phenotypes [169]. In addition, the salidroside derivative SHPL-49 selectively modulates excessive autophagy in microglia. SHPL-49 downregulates LAMP-2, impairs autophagosome–lysosome fusion, and leads to LC3-II and P62 accumulation in OGD-treated BV2 cells and pMCAO rats. These changes restore autophagic flux, suppress NF-κB signaling, lower IL-6, IL-1β, and iNOS levels, and improve neuronal viability in co-culture models—demonstrating its potential for precision autophagy modulation [152,153].

In addition to small molecules, peptide-based delivery systems have emerged as promising tools for modulating microglial autophagy with enhanced specificity. A novel cell-penetrating peptide, Tat-SIRT5-CTM, has been developed to selectively induce lysosomal degradation of SIRT5 in microglia, thereby mitigating microglia-driven neuroinflammation after ischemic stroke. By disrupting the SIRT5–ANXA1 interaction and facilitating SIRT5 clearance, this strategy enhances ANXA1 membrane localization and secretion, ultimately reducing infarct volume, preserving neuronal integrity, and improving neurological outcomes in stroke models—all without apparent toxicity [154]. Moreover, Tat-NTS—a related peptide—protects against cerebral ischemia/reperfusion injury by enhancing SUMOylation of ANXA1, thereby promoting NBR1-dependent selective autophagic degradation of IKKα, suppressing NF-κB activation, and reducing IL-1β and TNF-α release in microglia. This targeted peptide modulation of autophagy confers robust neuroprotection and behavioral improvement in rodent MCAO models [155].

Furthermore, exogenous transfer of functional mitochondria (F-Mito) isolated from bone marrow mesenchymal stem cells significantly improves neural cell survival post-ischemic stroke [170]. Notably, neurons and endothelial cells preferentially internalize F-Mito, which reduces intracellular ROS levels and enhances resilience to ischemia/reperfusion injury. Importantly, host cell ROS levels act as a gatekeeper—higher ROS promotes mitochondrial uptake, while ROS scavengers diminish it—revealing a self-regulating mechanism of targeted rescue. This strategy expands the therapeutic toolbox by offering a mitochondria-based intervention that complements autophagy-targeted approaches and warrants further exploration for its translational feasibility. While these agents exhibit favorable safety profiles, their lack of microglia-specific precision and context-dependent variability remain critical limitations.

Despite encouraging preclinical evidence, all pharmacological modulators of microglial autophagy or mitophagy in ischemic stroke remain at the preclinical stage, with none having progressed to formal clinical trials. Representative agents and their mechanisms—including β-elemene, ligustilide, Urolithin A, and SS-31—are summarized in Table 2. This translational gap highlights the urgent need for drug development strategies that move beyond mechanistic validation toward clinically viable approaches.

Drawing from recent preclinical evidence on timing-dependent autophagy modulation—such as early-phase activation via AMPK to enhance clearance of damaged mitochondria [100]—followed by later-phase suppression through agents like 3-MA in ischemic post-conditioning paradigms [171], future therapeutic strategies should move beyond vague “nuanced strategies” toward phase-specific, pathway-selective interventions. For example, an initial mitophagy boost may enhance mitochondrial resilience, followed by controlled suppression to prevent overactivation. Additionally, using mitophagy-targeted modulators such as Urolithin A—which improves mitochondrial quality without inducing broad autophagy—can further refine intervention precision. Such time-, dose-, and pathway-tailored regimens exemplify actionable and translationally viable approaches for microglial autophagy modulation. In this context, nattokinase—a serine protease derived from fermented soy—has demonstrated promising neuroprotective effects in tMCAO models by concurrently targeting oxidative stress, inflammation, and coagulation cascades. Its enzymatic activity mitigates ROS accumulation, suppresses inflammatory cytokines, reduces infarct volume, and improves neurological function, providing a unique multi-target therapeutic profile relevant to both microglial and neuronal injury [172].

### 3.4. Human Evidence and Biomarker Progress

While most mechanistic insights into microglial autophagy are derived from rodent or in vitro models, emerging evidence from human studies is beginning to bridge this gap.

Single-cell and spatial transcriptomics in human post-stroke tissue remain scarce, but preliminary analyses suggest that increased expression of autophagy-related genes such as ATG5 and PINK1 in microglial-enriched clusters correlates with larger infarct volumes and worse recovery outcomes. Supporting this, elevated serum ATG5 levels in acute ischemic stroke patients were associated with greater neurological deficits, recurrence, and mortality [146]. These findings, although limited in resolution, highlight both the promise and current insufficiency of autophagy-related gene programs as prognostic markers.

Circulating biomarkers also show promise. Serum ATG5 is dynamically elevated across multiple timepoints after stroke (admission through day 90), with higher levels predicting poor outcomes [146]. Likewise, circulating miR-124 exhibits a biphasic pattern—early decrease within 24 h followed by recovery over 48–72 h—which correlates with infarct volume and recovery trajectories [173]. Consistent with these findings, Zhou et al. (2021) reported that serum miR-124 is significantly downregulated in acute ischemic stroke patients and serves as both a diagnostic and prognostic marker [174]. However, these biomarkers are not microglia-specific, as they can also originate from neurons, astrocytes, and peripheral immune cells, which limits their interpretive precision for microglial autophagy. Validation of autophagy biomarkers in AIS patients should therefore include CSF measurements (e.g., Beclin-1, LC3B) and their correlation with infarct volume and neurological recovery metrics such as NIHSS and mRS [175].

Molecular imaging is another exploratory avenue. PET tracers such as radiolabeled DPA-714 (a TSPO ligand) have been applied in acute stroke patients, showing increased uptake in infarcted and perilesional regions [176]. Yet, TSPO primarily reflects microglial/macrophage activation rather than autophagic flux, and species-specific differences between rodents and humans further complicate interpretation [106,107]. Recent work further suggests that TSPO may not only serve as an imaging marker but also actively modulate microglial pathology by disrupting autophagic flux and impairing lysosomal degradation, thereby exacerbating inflammation after ischemia/reperfusion; inhibition of TSPO restored autophagic homeostasis and reduced ROS accumulation and cytokine release [177]. These findings highlight the need to distinguish between TSPO’s imaging utility and its mechanistic implications. To advance this field, future studies should evaluate novel tracers such as [^18F]PBR06 and experimental probes targeting lysosomal or autophagy-specific proteins, which may enable more direct visualization of autophagic or mitophagic flux in microglia [178,179].

Early-phase biomarker-guided clinical trials represent another critical priority. Designing pilot studies of mitophagy-enhancing agents (e.g., Urolithin A) with autophagy biomarkers as endpoints could help define optimal treatment windows, identify responsive subpopulations, and refine trial designs [180]. In parallel, human iPSC-derived microglia combined with multi-omics profiling (transcriptomics, proteomics, metabolomics) offer a transformative preclinical platform to resolve cell-specific and temporal regulation of autophagy under ischemic stress [181].

Collectively, current human evidence remains limited and largely indirect. Small sample sizes, reliance on post-mortem tissue, and absence of longitudinal monitoring hinder definitive conclusions. Furthermore, the commonly used autophagy markers LC3 and ATG5 are not microglia-specific, complicating mechanistic interpretation. Future progress will require integrated multi-omics platforms (single-cell transcriptomics, proteomics, metabolomics) coupled with imaging and clinical datasets to establish causal links between microglial autophagy and stroke outcomes.

### 3.5. Delivery Challenges and Future Directions

Even if mechanistic insights into microglial autophagy are increasingly refined, therapeutic translation faces a major barrier: efficient and cell-specific delivery across the blood–brain barrier (BBB).

Mesenchymal stem cell-derived extracellular vesicles (MSC-EVs) have shown promise as delivery vehicles capable of crossing the BBB and selectively targeting microglia. By transporting therapeutic miRNAs, proteins, or even mitochondria, MSC-EVs reduce ROS production and attenuate neuroinflammation in preclinical stroke models [182]. Similarly, nanocarrier systems such as PINK1 siRNA-loaded PLGA nanoparticles have achieved microglia-targeted delivery in rodent photothrombotic stroke, reducing excessive mitophagy, infarct volume, and improving functional outcomes [183]. These platforms demonstrate proof-of-concept that microglia-specific manipulation of autophagy is feasible.

Nonetheless, delivery efficiency, safety, and spatiotemporal precision remain unresolved challenges. Small-molecule drugs often fail to reach therapeutic concentrations within microglia due to poor BBB permeability, while systemic administration carries a risk of off-target effects. Engineered carriers—including MSC-EVs and biodegradable nanoparticles—partially overcome these barriers, but their clinical translation requires optimization to avoid immune activation and ensure reproducibility.

Future strategies will need to combine selective delivery technologies with precise temporal control of autophagy modulation. This will likely involve:Engineering carriers for enhanced BBB penetration and microglia specificity.Integrating spatiotemporal monitoring, using advanced imaging and biomarker profiling to guide dosing windows.Rigorous immune monitoring to mitigate the risk of autoimmunity, since dysregulated autophagy has been implicated in systemic inflammatory disorders [184,185].

Taken together, these innovations represent critical steps toward achieving safe and effective microglia-targeted therapies for ischemic stroke.

Beyond microglia, other glial populations also contribute to shaping the autophagic landscape in the ischemic brain. Astrocytes engage in extensive cross-talk with microglia via cytokines, chemokines, and extracellular vesicles, modulating inflammatory tone and potentially regulating microglial autophagy pathways [186,187]. Oligodendrocytes likewise rely on autophagy for myelin integrity and white matter repair; in models of chronic cerebral hypoperfusion, enhancing autophagy in oligodendrocytes improves oligodendrocyte survival and cognitive outcomes [188,189]. Although direct evidence linking astrocytic or oligodendrocytic autophagy to microglial autophagy during ischemic stroke is currently lacking, these findings underscore the importance of considering glial network interactions when interpreting autophagy-based interventions in the injured brain.

### 3.6. Clinical Outlook

Given that ischemia-induced mitochondrial dysfunction and maladaptive microglial activation are central drivers of neuronal injury, targeting autophagy and mitophagy represents a rational therapeutic strategy in ischemic stroke. Microglial mitophagy sits at the crossroads of pathophysiology and intervention, linking mitochondrial quality control with immune regulation. Building upon this mechanistic rationale, the following section outlines the current pharmacological landscape and translational barriers.

Despite encouraging preclinical evidence, all pharmacological modulators of microglial autophagy or mitophagy in ischemic stroke remain at the preclinical stage, with no agent having advanced to clinical trials. Representative compounds—including β-elemene, ligustilide, Urolithin A, and SS-31—show promising efficacy in animal models, yet translational hurdles persist. Notably, although Urolithin A has demonstrated safety and oral bioavailability in elderly humans, with favorable mitochondrial and anti-inflammatory biomarker profiles in randomized trials [190], no stroke-specific data exist.

Currently, the only FDA-approved pharmacologic intervention for acute ischemic stroke remains intravenous rt-PA, which is effective yet hemorrhagic-risk-limited [191,192], and mechanical thrombectomy—a standard of care for eligible patients [193]. Microglial autophagy modulators, by contrast, would likely serve as adjuncts to reperfusion therapy, targeting downstream immunometabolic injury.

The translational gap in ischemic stroke reflects broader challenges in mechanism–delivery–population alignment. Effective strategies must address mechanistic complexity of microglial responses, BBB penetration and cell-type specificity, and development of human-relevant biomarkers and imaging endpoints. Next-generation approaches should therefore combine pathway-selective modulators with precision delivery systems (e.g., engineered extracellular vesicles or nanocarriers) and incorporate microglia-derived biomarkers for patient stratification.

In addition to their role in neuroinflammation, microglia are emerging as key players in post-stroke regeneration. Enhanced mitophagy may suppress chronic inflammation and indirectly support angiogenesis and neurogenesis, although direct causation remains unproven. Future studies integrating microglia-specific autophagy modulation with regenerative outcome measures may clarify whether these pathways can be harnessed to promote repair [5,156].

Comparative studies in other neurological disorders underscore similar translational challenges. In Parkinson’s disease, autophagy activation via agents such as metformin has shown neuroprotective effects by activating the AMPK/ULK1/PINK1/Parkin pathway, thereby enhancing autophagy and mitophagy, promoting mitochondrial clearance, and reducing neuronal apoptosis in experimental models [112,151]. Trehalose, another autophagy enhancer, facilitates aggregate clearance and prevents protein misfolding, and has shown benefits in animal models of neurodegeneration, including PD [194]. In Alzheimer’s disease, mitophagy inducers reduce amyloid-β and tau accumulation while partially reversing cognitive deficits in experimental systems [195]. In multiple sclerosis, autophagy exerts a dual role: protective against oxidative stress and inflammation, yet potentially detrimental when overactivated, thereby exacerbating neuropathology [196,197]. Together, these parallels highlight the double-edged nature of autophagy—protective in acute injury, but potentially harmful under conditions of chronic dysregulation.

At the same time, while these strategies highlight therapeutic promise, it is essential to consider potential risks of chronic autophagy modulation. Dysregulated autophagy is implicated in autoimmune and inflammatory disorders (e.g., systemic lupus erythematosus, rheumatoid arthritis, inflammatory bowel disease, multiple sclerosis). Genetic variants in autophagy-related genes (such as ATG16L1, ATG5) increase susceptibility to such conditions, and both excessive inhibition and activation of autophagy can disrupt immune tolerance via altered antigen presentation, cytokine imbalance, and lymphocyte dysregulation [184,185,198]. Therefore, future translational research should incorporate comprehensive immune monitoring—including autoantibody panels, lymphocyte activity markers, and systemic inflammation biomarkers—to mitigate autoimmune risks.

## 4. Conclusions

Microglial autophagy—particularly mitophagy—functions as a context-dependent regulator in ischemic stroke, capable of sustaining mitochondrial quality control and promoting anti-inflammatory polarization when appropriately activated, but also contributing to neurotoxicity when dysregulated or excessive. This duality underscores the importance of temporal and spatial precision in therapeutic modulation.

Evidence to date is dominated by preclinical models, whereas human data remain scarce and largely indirect. Collectively, this scoping review highlights both the therapeutic promise and translational challenges of targeting microglial autophagy in ischemic stroke. Advances in single-cell transcriptomics, peripheral biomarkers, and exploratory imaging provide early translational insights, yet substantial gaps persist—including the absence of longitudinal human datasets, limited cell-type–specific resolution, and insufficient clinical validation of pharmacological modulators.

Future research must define temporal windows for safe and effective autophagy modulation in human stroke, establish reliable microglia-specific biomarkers, and evaluate translational platforms such as human iPSC-derived microglia. Additional priorities include overcoming blood–brain barrier delivery constraints, minimizing off-target immune effects such as autoimmunity, and assessing whether combined strategies—such as pairing autophagy modulators with mitochondrial protectants or anti-inflammatory agents—can yield synergistic benefit. Addressing these gaps will be essential to move microglial autophagy modulation from experimental promise toward clinically relevant stroke therapy.

## Figures and Tables

**Figure 1 biology-14-01269-f001:**
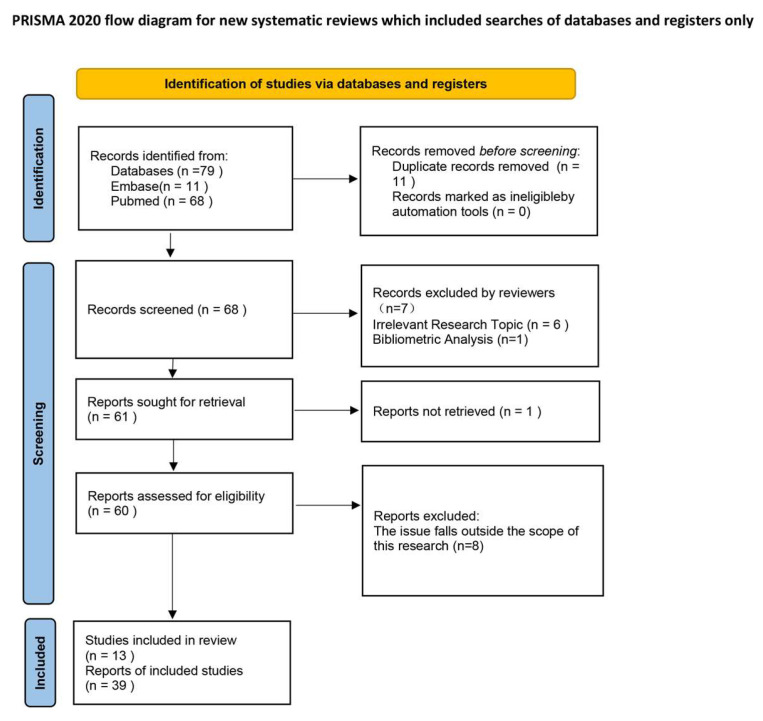
PRISMA flow diagram illustrating the study selection process.

**Figure 2 biology-14-01269-f002:**
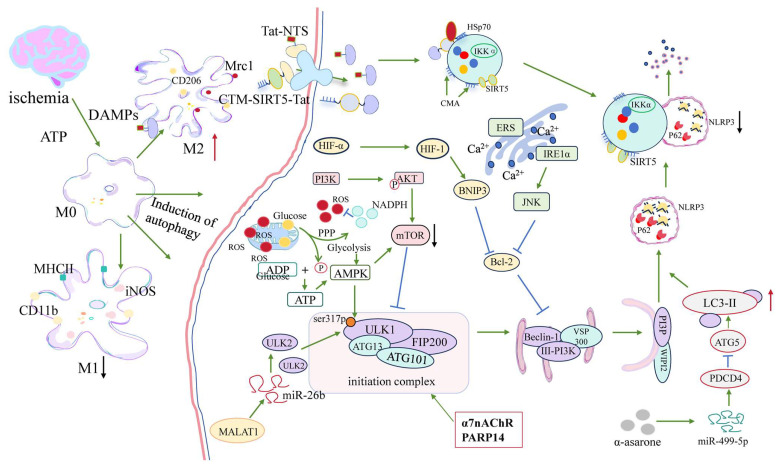
Protective regulatory networks of microglial autophagy following ischemic stroke. After cerebral ischemia, ATP depletion and DAMP release trigger microglial activation and polarization. This figure summarizes the molecular signaling pathways that promote autophagy-mediated neuroprotection in microglia. Stress signals activate the AMPK–ULK1–Beclin1 complex and inhibit mTOR, thereby inducing autophagy. ER stress–mediated IRE1/JNK and HIF-1α–BNIP3 pathways also contribute to autophagy and mitophagy. Key upstream regulators such as α7nAChR and PARP14 modulate inflammatory states and enhance autophagic flux. Non-coding RNAs—including MALAT1, miR-26b, and miR-499-5p—participate in ceRNA networks that facilitate ULK2 and ATG5 expression. Pharmacological agents such as α-asarone promote protective autophagy by targeting these axes. Through mitochondrial quality control, inflammasome suppression, and phenotypic transition from M1 to M2, autophagy acts as a critical mediator of microglial homeostasis in the early phase post-stroke. In this figure, green arrows indicate activation/promotion, blue arrows indicate inhibition, red arrows indicate increased expression, and black arrows indicate reduced expression.

**Figure 3 biology-14-01269-f003:**
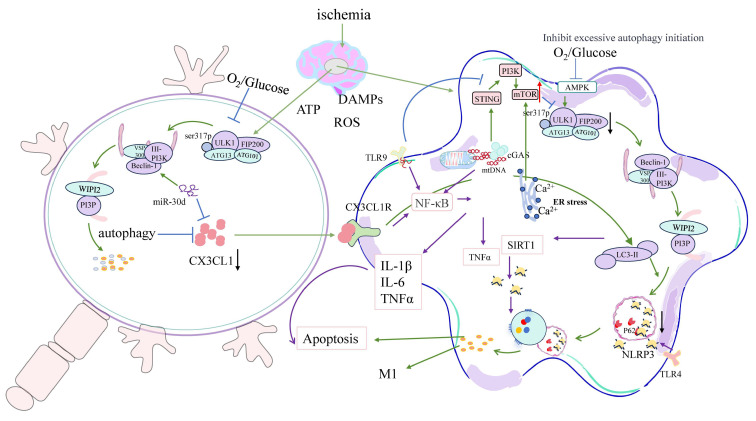
Pathological signaling pathways underlying dysregulated microglial autophagy after ischemic stroke. Following cerebral ischemia, ATP depletion and ROS overproduction lead to mitochondrial dysfunction and the release of DAMPs and mtDNA, which trigger innate immune responses in microglia. This figure summarizes the molecular mechanisms by which sustained or excessive autophagy contributes to neuroinflammation and neuronal injury. Activation of TLR9 and the cGAS–STING axis promotes NF-κB–mediated expression of IL-1β, IL-6, and TNF-α. Mitochondrial Ca^2+^ overload and ER stress exacerbate autophagic activity via Beclin1–PI3K complexes, while persistent p62 accumulation amplifies NLRP3 inflammasome activation. Disruption of CX3CL1–CX3CR1 signaling impairs neuron–microglia communication and further skews microglia toward an M1-like pro-inflammatory phenotype. Additionally, miR-30d suppresses CX3CL1 expression, contributing to inflammatory amplification and apoptotic signaling. These maladaptive pathways form a positive feedback loop linking defective autophagic resolution, mitochondrial DAMP signaling, and chronic neuroinflammation in ischemic stroke. In this figure, green arrows indicate activation/promotion, blue arrows indicate inhibition, black arrows represent downregulation or suppression, red arrows indicate upregulation or enhancement, and purple arrows represent inflammatory signaling pathways.

**Figure 4 biology-14-01269-f004:**
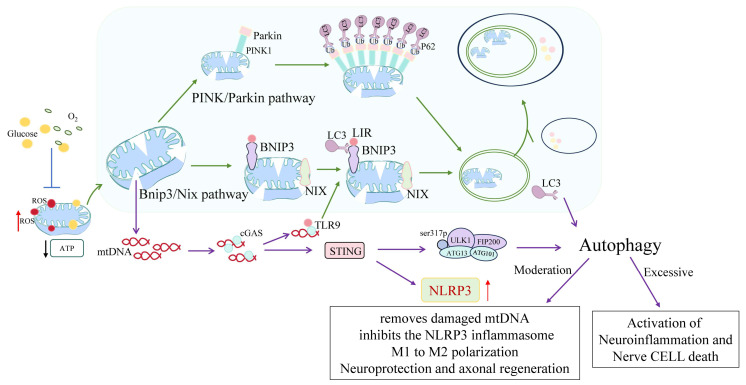
Molecular mechanisms underlying microglial mitophagy in ischemic stroke and their immunometabolic implications. Upon ischemic injury, mitochondrial depolarization and ROS accumulation initiate selective mitophagy through two primary pathways: PINK1/Parkin and BNIP3/NIX. In the PINK1/Parkin axis, damaged mitochondria accumulate ubiquitinated proteins on their outer membranes, which subsequently recruit autophagy receptors such as p62 to facilitate autophagosome formation. In parallel, BNIP3 and NIX act as mitophagy receptors by directly interacting with LC3 via their LC3-interacting region (LIR) motifs, thus mediating receptor-dependent mitophagy. Mitochondrial DNA (mtDNA) leakage from compromised organelles activates the cGAS–STING signaling cascade, which in turn stimulates the NLRP3 inflammasome, escalating neuroinflammation. Moderate levels of mitophagy counteract this process by promoting M1-to-M2 microglial polarization, thereby enhancing neuroprotection and axonal regeneration. Conversely, excessive or insufficient mitophagy exacerbates neuroinflammatory responses and contributes to neuronal cell death. This figure encapsulates the dual immunometabolic outcomes of mitophagy modulation in microglial responses to cerebral ischemia. In this figure, green arrows represent activation or promotion, blue arrows indicate inhibition, black arrows denote decreased activity or expression, red arrows reflect increased expression, and purple arrows indicate autophagy-related pathways.

**Table 1 biology-14-01269-t001:** Dual roles of microglia and their regulation by autophagy/mitophagy in ischemic stroke.

Role	Mechanisms/Features	Functional Outcomes	References
Neuroprotective (adaptive/M2-like/reparative)	Autophagy/mitophagy clears damaged mitochondria and protein aggregates Suppression of ROS and NLRP3 inflammasome activation Release of anti-inflammatory cytokines (IL-4, IL-10, TGF-β) Enhanced phagocytosis of debris and synaptic pruning of irreversibly damaged neurons Promotion of angiogenesis and remyelination (via VEGF, BDNF) Crosstalk with astrocytes and oligodendrocytes supports repair	Reduced oxidative stress Maintenance of mitochondrial quality control Enhanced neuronal survival and synaptic recovery Improved neurological outcomes	[15,102,103,104]
Neurotoxic (maladaptive/M1-like/inflammatory)	Excessive or defective autophagy impairs lysosomal flux Release of pro-inflammatory mediators (TNF-α, IL-1β, IL-6, NO, ROS) Overactivation of NLRP3 inflammasome Aberrant synaptic pruning of viable neurons BBB breakdown via MMP upregulation Crosstalk with astrocytes exacerbates neuroinflammation and excitotoxicity	Exacerbated neuroinflammation and secondary neuronal death BBB disruption and edema Chronic gliosis and cognitive decline Worse stroke prognosis	[105,106,107,108]

**Table 2 biology-14-01269-t002:** Key molecular regulators of microglial mitophagy in ischemic stroke.

Molecule/Regulator	Pathway/Mechanism	Effect on Microglial Autophagy/Mitophagy	Stroke Phase	References
PINK1/Parkin	Ubiquitin–proteasome signaling; mitochondrial depolarization sensor	Induces mitophagy, clears damaged mitochondria, reduces ROS	Acute/Subacute	[12,42,112,113]
BNIP3/NIX	Hypoxia-inducible, LC3-interacting region (LIR) receptor	Promotes receptor-mediated mitophagy, suppresses inflammasome activation	Acute/Subacute	[45,46,47,120]
FUNDC1	Mitochondrial outer membrane receptor; LIR motif	Promotes mitophagy under hypoxia/ischemia, regulates mitochondrial quality control	Acute	[114,120,121]
HIF-1α	Hypoxia-inducible factor, transcriptional activation of BNIP3/NIX	Indirectly enhances mitophagy under ischemia/hypoxia	Acute	[45,46,47,120]
Beclin-1	Class III PI3K complex	Initiates autophagosome formation, regulates autophagy flux	Acute/Subacute	[46,47,122]
PGC-1α/ULK1	Mitochondrial biogenesis coactivator; ULK1 initiates autophagy	PGC-1α enhances ULK1-dependent mitophagy; shifts microglia toward M2 phenotype	Subacute/Recovery	[116]
MALAT1 (lncRNA)	Competes with miR-30d/ULK2 axis	Promotes autophagy by derepressing ULK2, regulates polarization	Acute/Subacute	[55,122]
TUG1 (lncRNA)	Regulates PINK1/Parkin pathway	Enhances mitophagy and protects against ischemic injury	Acute/Subacute	[57]
miR-26b	Targets ULK2	Suppresses autophagy, promotes pro-inflammatory activation	Acute	[55]
miR-30a	Targets Beclin-1	Inhibits autophagy, exacerbates ischemic injury	Acute	[122]
miR-124	Targets STAT3, modulates inflammatory response	Enhances autophagy, promotes M2-like phenotype	Subacute/Recovery	[121,122]
Exosomal miRNAs (miR-126, miR-223)	Delivered via MSC- or exercise-derived exosomes	Modulate autophagy/mitophagy, reduce neuroinflammation	Subacute/Recovery	[123,124]
DRP1	Mitochondrial fission mediator	Excessive activation induces fragmentation and autophagic stress	Acute	[125,126]
MFN2/OPA1	Mitochondrial fusion proteins	Promote mitochondrial integrity, restrain excessive mitophagy	Subacute/Recovery	[127]

**Table 3 biology-14-01269-t003:** Pharmacological agents modulating microglial autophagy/mitophagy in ischemic stroke.

Compound	Target Pathway/Mechanism	Effect on Microglial Autophagy/Mitophagy	Key Findings (Preclinical Model)	Development Stage	References
β-Elemene	AKT/mTOR	Promotes autophagy and shifts microglia toward M2 phenotype	Reduces infarct size and neuroinflammation in MCAO mice	Preclinical (rodent models)	[10]
Baicalein	PI3K/Akt/mTOR	Inhibits autophagy, reduces neuronal apoptosis via anti-autophagic signaling	Promotes PI3K/Akt/mTOR activation, reduces LC3-II/LC3-I ratio, protects neurons from I/R injury	Preclinical	[97]
Salidroside	AMPK/TSC2/mTOR	Activates autophagy, reduces oxidative stress	Neuroprotection in ischemia/reperfusion (I/R) models	Preclinical	[147]
Ursolic Acid	PPARγ–MMP2 regulation; antioxidant via Nrf2/HO-1	Indirect modulation of autophagy via microglial polarization and ECM balance	Promotes microglial M2 polarization via PPARγ–MMP2; restores MMP/TIMP balance and reduces neuronal injury	Preclinical	[148,149,150]
PTP1B inhibitor	PERK/ER stress–autophagy axis	Regulates microglial autophagy and inflammatory signaling	Reduces ischemic neuronal death and neuroinflammation	Preclinical	[20]
Metformin	AMPK activation	Promotes autophagy/mitophagy, enhances mitochondrial clearance	Neuroprotection in stroke and neurodegeneration models	Approved drug (diabetes); preclinical in stroke	[112,151]
STS (Sodium Tanshinone IIA Sulfonate)	PP2A-mediated enhancement of autophagy in microglia	Promotes autophagic flux and exerts anti-inflammatory effects	Restores mitochondrial function and reduces neuronal apoptosis in OGD/R models; increases Beclin-1 and ATG5 expression, decreases p62, upregulates IL-10/TGF-β/BDNF, while inhibiting IL-1β/IL-2/TNF-α	Preclinical	[117]
Ligustilide	PINK1/Parkin-dependent mitophagy; BNIP3–LC3-mediated mitophagy; PI3K/Akt-mediated anti-apoptotic signaling	Enhances mitophagy, reduces oxidative stress and apoptosis	Improves mitochondrial function and neuronal survival via mitophagy in MCAO/R models; attenuates ROS and apoptosis via PI3K/Akt; preserves BBB integrity in OGD models	Preclinical (rodent, in vitro models)	[135,136,137]
Urolithin A	Activates autophagy and suppresses ER stress; mitophagy observed in non-CNS models	Enhances autophagy, reduces neuronal apoptosis and inflammation	Protects against ischemic injury; mitophagy demonstrated in muscular dystrophy models	Phase I safety trial completed; no stroke-specific trials	[101,141]
SS-31 (Elamipretide)	Cardiolipin binding; mitochondrial stabilization; modulation of AKT/mTOR	Restores autophagic flux and improves mitochondrial function	Demonstrates neuroprotection in traumatic brain injury (TBI) and non-CNS ischemia models by reversing mitochondrial dysfunction; limited direct stroke evidence	Clinical (other diseases, incl. mitochondrial myopathy); none in stroke	[140,142]
SHPL-49 (Salidroside derivative)	LAMP-2/autophagosome–lysosome fusion	Inhibits excessive autophagy and restores flux	Reduces infarct size and inflammation in I/R models	Preclinical	[152,153]
Tat-SIRT5-CTM (peptide)	SIRT5 degradation/ANXA1 pathway	Modulates microglial autophagy, enhances anti-inflammatory response	Improves ischemic outcomes in rodent models	Preclinical (proof-of-concept)	[154]
Tat-NTS peptide	Enhances ANXA1 SUMOylation → NBR1-mediated selective autophagic degradation of IKKα, suppressing NF-κB	Modulates microglial selective autophagy and shifts phenotype toward anti-inflammation	Reduces infarct volume and improves neurobehavioral recovery in MCAO mice via microglial autophagy modulation	Preclinical (proof-of-concept)	[155]

## Data Availability

No new data were created or analyzed in this study. Data sharing is not applicable to this article.

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
