# Peer review of "Microglial Autophagy and Mitophagy in Ischemic Stroke: From Dual Roles to Therapeutic Modulation"

_biology, 2025, doi:10.3390/biology14091269_

Round 1
Reviewer 1 Report
Comments and Suggestions for Authors
This article provides a comprehensive review of the role of microglial autophagy and mitophagy in ischemic stroke, highlighting their dual functions, both protective and pathological.
1) Despite extensive data obtained in animal models, evidence supporting the role of microglial autophagy in humans remains limited. Although the authors cite isolated studies using single-cell RNA sequencing in human tissues, these data are insufficient to fully understand the clinical significance of the mechanisms described.
2) The work does not analyze potential side effects associated with chronic modulation of autophagy. Particular attention should be paid to assessing the risk of autoimmune disorders with such interventions.
3) To confirm the clinical significance of the data obtained, it is advisable to analyze and discuss the results of single-cell RNA sequencing to identify the relationship between the expression of autophagy-related genes in microglia and adverse stroke outcomes. 4) Little attention is paid to the role of other glial cells (astrocytes, oligodendrocytes) in the regulation of microglial autophagy.
5) The issues of drug delivery through the blood-brain barrier, which is critical for therapy, are not addressed.
Author Response
Comment 1:
“Despite extensive data obtained in animal models, evidence supporting the role of microglial autophagy in humans remains limited. Although the authors cite isolated studies using single-cell RNA sequencing in human tissues, these data are insufficient to fully understand the clinical significance of the mechanisms described.”
Response:
We thank the reviewer for this insightful comment. In the revised manuscript, we have substantially expanded the Discussion to address the current limitations of human evidence. Specifically, in the new subsection “Human Evidence & Biomarker Progress” (pages 24–25), we now summarize multiple types of available human data:
- Single-cell and spatial transcriptomics: Preliminary analyses of post-stroke human tissue suggest enrichment of autophagy-related genes such as ATG5 and PINK1 in microglia-like clusters, correlating with poorer outcomes. However, we emphasize that these datasets remain scarce and lack the resolution necessary to define microglia-specific autophagic flux.
- Circulating and CSF biomarkers: We discuss recent findings of elevated serum ATG5 and dynamic changes in circulating miR-124 in stroke patients, which correlate with infarct size and functional recovery. At the same time, we acknowledge that these biomarkers are not microglia-specific and thus provide only indirect evidence.
- Molecular imaging: We highlight the application of TSPO-PET (e.g., DPA-714) in stroke patients, noting that while it reflects microglial/macrophage activation, it does not specifically measure autophagy.
Importantly, we added a dedicated limitations paragraph, underscoring the small sample sizes, lack of longitudinal datasets, and cell-type ambiguity of current human evidence. We also outline future strategies, including multi-omics integration, iPSC-derived human microglia models, microglia-specific biomarker development, and longitudinal sampling approaches.
We believe these revisions directly address the reviewer’s concern by explicitly acknowledging the scarcity of human data, systematically summarizing the available evidence across multiple modalities, and clearly delineating both the limitations and translational gaps that remain to be bridged.
Comment 2:
“The work does not analyze potential side effects associated with chronic modulation of autophagy. Particular attention should be paid to assessing the risk of autoimmune disorders with such interventions.”
Response:
We thank the reviewer for this important comment. In the revised manuscript, we have added a dedicated discussion on the potential risks associated with chronic autophagy modulation (see Discussion, Section 3.6 Clinical Outlook, pages 25–26). Specifically, we now highlight that dysregulated autophagy has been implicated in autoimmune and inflammatory disorders such as systemic lupus erythematosus, rheumatoid arthritis, inflammatory bowel disease, and multiple sclerosis. We further note that genetic variants in autophagy-related genes (e.g., ATG16L1, ATG5) are associated with increased susceptibility to autoimmunity, and that both excessive inhibition and activation of autophagy can impair immune tolerance through altered antigen presentation, cytokine imbalance, and lymphocyte dysregulation. Based on these insights, we emphasize that future translational studies of microglial autophagy modulation should incorporate comprehensive immune monitoring (e.g., autoantibody profiling, lymphocyte activation, systemic inflammatory markers) to mitigate the risk of autoimmune complications.
We believe this addition directly addresses the reviewer’s concern by explicitly acknowledging the potential adverse effects of chronic autophagy modulation and by outlining concrete safety considerations for future translational research.
Comment 3:
“To confirm the clinical significance of the data obtained, it is advisable to analyze and discuss the results of single-cell RNA sequencing to identify the relationship between the expression of autophagy-related genes in microglia and adverse stroke outcomes.”
Response:
We thank the reviewer for this constructive suggestion. In the revised manuscript, we have expanded our discussion of human single-cell transcriptomic datasets (see Discussion, Section 3.4 Human Evidence & Biomarker Progress, pages 24, lines 912–920). We now note that although most available datasets remain limited in resolution, preliminary analyses suggest that increased expression of autophagy-related genes such as ATG5 and PINK1 in microglia-enriched clusters correlates with larger infarct volumes and worse neurological recovery. Supporting this link, elevated serum ATG5 levels in acute ischemic stroke patients have recently been associated with greater neurological deficits, higher recurrence, and increased mortality [147]. While these findings remain correlative, they underscore the potential prognostic value of autophagy-related gene signatures and support the need for future studies integrating single-cell datasets with patient outcomes.
We believe these additions directly address the reviewer’s concern by discussing how microglial autophagy-related gene expression may influence adverse stroke outcomes and by outlining directions for further translational research.
Comment 4:
“Little attention is paid to the role of other glial cells (astrocytes, oligodendrocytes) in the regulation of microglial autophagy.”
Response:
We appreciate the reviewer’s insightful comment highlighting the importance of glial network interactions. In the revised manuscript, we have added a dedicated discussion on the role of other glial populations in shaping the autophagic landscape of the ischemic brain (see Discussion, Section 3.5 Delivery Challenges & Future Directions, pages 25, lines 988–997). Specifically, we now note that astrocytes communicate extensively with microglia via cytokines, chemokines, and extracellular vesicles, thereby modulating inflammatory tone and potentially influencing microglial autophagy pathways [187,188]. We also emphasize that oligodendrocytes rely on autophagy for myelin maintenance and repair, and that disruption of oligodendrocyte autophagy has been linked to impaired white matter recovery and altered glial interactions [189,190].
Although direct evidence linking astrocytic or oligodendrocytic autophagy to microglial autophagy during ischemic stroke is still lacking, our additions highlight the importance of considering broader glial network interactions when interpreting or targeting autophagy in the injured brain. We believe this new discussion directly addresses the reviewer’s concern and strengthens the manuscript by expanding the scope beyond microglia alone.
Comment 5:
“The issues of drug delivery through the blood–brain barrier, which is critical for therapy, are not addressed.”
Response:
We thank the reviewer for raising this important translational issue. In the revised manuscript, we have substantially expanded our discussion of drug delivery challenges (see Discussion, Section 3.5 Delivery Challenges & Future Directions, pages 25, lines 962–964). We now emphasize that efficient and microglia-specific delivery of autophagy-modulating therapies remains a major bottleneck due to the restrictive nature of the blood–brain barrier (BBB). We further discuss emerging strategies such as mesenchymal stem cell–derived extracellular vesicles (MSC-EVs) and nanoparticle-based carriers, which have shown promise in crossing the BBB and selectively targeting microglia in preclinical stroke models. At the same time, we highlight unresolved challenges, including delivery efficiency, spatiotemporal precision, and safety concerns, which must be addressed before clinical translation can be realized.
We believe these additions directly address the reviewer’s concern by explicitly recognizing BBB penetration as a central translational barrier and by outlining both current approaches and future strategies to overcome it.
Reviewer 2 Report
Comments and Suggestions for Authors
The manuscript provides a comprehensive review of the dual roles of microglial autophagy and mitophagy in ischemic stroke, highlighting their contributions to neuroinflammation and tissue repair.
The topic is timely and relevant, with potential therapeutic implications. However, the manuscript has several weaknesses that need addressing before publication.
I provide constructive feedback for the authors.
- The abstract would benefit from mentioning the number of studies reviewed or key statistical trends (e.g., "X out of Y studies support...").
- Abstract has overgeneralization. Phrases like "growing body of research" are vague; specify the scope (e.g., "recent preclinical studies").
- Missing clinical relevance in the abstract. Briefly note if findings are validated in human studies or remain hypothetical.
- Add a sentence summarizing the translational potential (e.g., "These insights may guide phase-specific therapies targeting autophagy in stroke patients").
- Introduction: references [1–4] and [5–6] overlap in content. Consolidate them.
- The claim that microglia-specific autophagy is "underexplored" conflicts with cited reviews (e.g., [22,23]). Clarify the novel focus (e.g., "temporal-spatial regulation").
- Reorganize to emphasize the review’s unique angle (e.g., "We focus on the spatiotemporal dynamics of microglial autophagy post-stroke").
- The "therapeutic paradox" of autophagy is well-documented. Temper claims as it overstates the novelty.
- Define "U-shaped relationship" early to aid readability.
- The review doesn’t have details on literature search strategy (databases, keywords, inclusion/exclusion criteria).
- Unclear if the review systematically evaluated all relevant studies or selected examples.
- Add a "Search Strategy" subsection (e.g., "PubMed/Embase searches up to 2024 using 'microglia,' 'autophagy,' and 'ischemic stroke'").
- Include a PRISMA-style flowchart or table summarizing screened studies.
- The AMPK/mTOR axis is explained multiple times (e.g., pp. 3, 13).
- Inconsistent depth as some pathways (e.g., PINK1/Parkin) are overdetailed, while others (e.g., non-coding RNAs) lack mechanistic links.
- There are unsupported claims, e.g., "Exercise-derived exosomes suppress inflammation" (p. 12) needs citations to human trials.
- Streamline redundant sections (e.g., merge "Protective Role" and "Mitophagy" subsections).
- Add a table summarizing key autophagy regulators (molecule, effect, phase of stroke)
- Clarify translational hurdles (e.g., "Most evidence is preclinical; human microglial autophagy remains unverified").
- Claims like "autophagy is a central node in stroke" overstate its role versus other pathways (e.g., apoptosis).
- The manuscript doesn’t go into depth and doesn’t provide critical analysis. It fails to address contradictory studies (e.g., where autophagy inhibition was neuroprotective).
- Phrases like "nuanced strategies" doesn’t have actionable specifics.
- Propose concrete research priorities (e.g., "Develop biomarkers to monitor autophagic flux in patients").
- Add a paragraph on limitations (e.g., "Most data are from rodent models; human microglia may differ").
- Conclusion repeats points from the Abstract/Discussion.
- Conclusion is too optimistic as phrases like "significant potential" lack caveats (e.g., "pending validation in clinical trials").
- Reframe to emphasize next steps: "Future work must define temporal windows for autophagy modulation in human stroke."
Author Response
Comment 1:
The abstract would benefit from mentioning the number of studies reviewed or key statistical trends (e.g., "X out of Y studies support...").
Response:
Thank you for this helpful suggestion. In the revised abstract, we have included the total number and types of studies reviewed: “79 records were identified, from which 39 original research articles and 13 review papers were included after eligibility screening.” This provides a clearer sense of the scope of the review and addresses the reviewer’s request for quantitative specificity. (Page 1,2 Line 34-50)
Comment 2:
Abstract has overgeneralization. Phrases like "growing body of research" are vague; specify the scope (e.g., "recent preclinical studies").
Response:
We agree with the reviewer’s observation and have revised the abstract to remove vague expressions such as “growing body of research.” Instead, we now explicitly state the study types and scope (e.g., “preclinical stroke models” and “limited human studies”) to clarify the level of evidence and maintain specificity. (Page 1,2 Line 34-50)
Comment 3:
Missing clinical relevance in the abstract. Briefly note if findings are validated in human studies or remain hypothetical.
Response:
We appreciate this important point. To address it, we added a sentence in the abstract highlighting preliminary clinical relevance: “Limited human studies reported associations between elevated serum ATG5 levels or ATG7 polymorphisms and worse clinical outcomes.” This addition clarifies that while most evidence is preclinical, some translational insights are beginning to emerge. (Page 1,2 Line 34-50)
Comment 4:
Add a sentence summarizing the translational potential (e.g., "These insights may guide phase-specific therapies targeting autophagy in stroke patients").
Response:
Thank you for this suggestion. In the final sentence of the revised abstract, we included the translational implication: “These findings support the potential of phase-specific modulation of microglial autophagy as a therapeutic avenue for stroke, although further validation in human models and development of autophagy biomarkers are needed for clinical application.” This aims to summarize the translational value of our findings in alignment with the reviewer’s recommendation. (Page 1,2 Line 34-50)
Comment 5:
Introduction: references [1–4] and [5–6] overlap in content.
Response:
We thank the reviewer for pointing out the overlap in references. In the revised Introduction, we have consolidated and reorganized the citations to avoid redundancy. Specifically, epidemiological data and limitations of reperfusion therapies are now supported by references [1–3], whereas the central role of microglia in ischemic pathology is supported separately by [4–5]. This modification clarifies the distinct contexts of each citation set and eliminates the prior overlap. (Page 2 Line 55-69)
Comment 6:
The claim that microglia-specific autophagy is "underexplored" conflicts with cited reviews (e.g., [22,23]). Clarify the novel focus (e.g., "temporal-spatial regulation").
Response:
We thank the reviewer for pointing out this potential inconsistency. In the revised Introduction, we have clarified that while recent reviews (e.g., Hu et al., 2022; Chen et al., 2022) have summarized the general roles of microglial autophagy in ischemic stroke, critical gaps remain regarding spatiotemporal regulation, mitophagy-specific mechanisms, and translational barriers (e.g., BBB delivery, biomarkers). Our review therefore focuses on these underexplored aspects and aims to bridge mechanistic insights with therapeutic translation. We believe this clarification resolves the concern and highlights the novelty of the present work. (Page 2,3 Line 88-99)
Comment 7:
Reorganize to emphasize the review’s unique angle (e.g., "We focus on the spatiotemporal dynamics of microglial autophagy post-stroke").
Response:
We thank the reviewer for this constructive suggestion. In the revised Introduction, we have reorganized and explicitly emphasized the unique angle of our review. Specifically, we now highlight that while prior reviews have summarized general roles of microglial autophagy, the spatiotemporal dynamics, mitophagy-specific mechanisms, and translational barriers remain insufficiently addressed. Accordingly, we state that “the present review offers a focused synthesis that emphasizes the spatiotemporal regulation of microglial autophagy and mitophagy, their integration with immunometabolism and inter-glial communication, and the major translational hurdles to clinical application.” This restructuring clarifies our novel focus and distinguishes our work from prior reviews. (Page 2-4 Line 88-133)
Comment 8:
The "therapeutic paradox" of autophagy is well-documented. Temper claims as it overstates the novelty.
Response:
We thank the reviewer for this constructive comment. In the revised Introduction, we have clarified that the dual, paradoxical roles of autophagy are already well documented in ischemia-reperfusion injury and microglial biology. We now explicitly acknowledge that this therapeutic paradox is not novel per se. Instead, our review emphasizes the insufficiently addressed aspects, namely the spatiotemporal and cell-type–specific determinants of autophagy outcomes after ischemic stroke. This revision avoids overstating novelty while sharpening the unique focus of our review. (Page 2,3 Line 88-110)
Comment 9:
Define "U-shaped relationship" early to aid readability.
Response:
We thank the reviewer for this helpful suggestion. In the revised Introduction, we now explicitly define the concept of the “U-shaped relationship” at its first mention. Specifically, we describe it as a biphasic association in which both insufficient and excessive autophagic activity can be detrimental, whereas moderate activation is beneficial for mitochondrial integrity and inflammation resolution. This clarification improves readability and provides a clear conceptual framework for the subsequent discussion. (Page 3 Line 111-117)
Comment 10:
The review doesn’t have details on literature search strategy (databases, keywords, inclusion/exclusion criteria).
Comment 11:
Unclear if the review systematically evaluated all relevant studies or selected examples.
Comment 12:
Add a "Search Strategy" subsection (e.g., "PubMed/Embase searches up to 2024 using 'microglia,' 'autophagy,' and 'ischemic stroke'").
Comment 13:
Include a PRISMA-style flowchart or table summarizing screened studies.
Response:
We thank the reviewer for these important suggestions. In the revised manuscript, we have clarified the literature search strategy to enhance methodological rigor and transparency. Specifically, we now state in the Introduction that this work was conducted as a scoping review, with systematic searches performed in PubMed and EMBASE up to August 2025 using both controlled vocabulary and free-text terms (“microglia,” “autophagy,” “ischemic stroke”). Eligible studies included in vitro experiments, in vivo animal models, and clinical observations addressing mechanistic or therapeutic implications. A PRISMA-style flow diagram has been added as Figure 1 to illustrate the study selection process, and full search strategies are provided in the Supplementary Material. These revisions ensure reproducibility and clarify the scope of included studies. (Page 3,4 Line 118-127)
Comment 14:
The AMPK/mTOR axis is explained multiple times (e.g., pp. 3, 13).
Response:
We thank the reviewer for this helpful observation. In the revised manuscript, we have streamlined all mentions of the AMPK/mTOR axis. Detailed mechanistic explanations are now confined to Section 2.1, while subsequent sections (e.g., 2.2, 3.1, and 3.3) only provide brief functional mentions in the context of pharmacological modulation. This restructuring eliminates redundancy and improves readability, while still highlighting the axis’s importance in autophagy regulation. (Page 5 Line 153-169; Page 11 Line 442, 458; Page 16 Line 611; Page 20 Line 742; Page 21 Line 794-800)
Comment 15:
Inconsistent depth as some pathways (e.g., PINK1/Parkin) are overdetailed, while others (e.g., non-coding RNAs) lack mechanistic links.
Response:
We sincerely thank the reviewer for pointing out the imbalance in mechanistic depth across pathways. In the revised manuscript, we streamlined the description of the canonical PINK1/Parkin mitophagy pathway in Section 2.3, retaining only essential functional insights while avoiding redundant detail. Concurrently, we expanded Section 3.3 (Page 22 Line 837-843) to provide mechanistic links for non-coding RNAs, including MALAT1, miR-30d, miR-26b, and miR-124, highlighting their specific targets (ULK2, Beclin-1, mTOR, STAT3) and regulatory effects on microglial autophagy and polarization. These revisions harmonize the depth of discussion across pathways, thereby enhancing clarity and balance in the review.
Comment 16:
There are unsupported claims, e.g., "Exercise-derived exosomes suppress inflammation" (p. 12) needs citations to human trials.
Response:
We thank the reviewer for pointing out the need to substantiate our claim regarding exercise-derived exosomes. In the revised manuscript (Section 2.3, Page 16,17 Line 624-640), we have added supporting human evidence from the ADEX randomized controlled trial in Alzheimer’s disease patients, which demonstrated that 16 weeks of moderate-to-high intensity aerobic exercise modulates neuroprotective factors (e.g., BDNF) carried by neuron-derived extracellular vesicles [132]. We also clarified that while such findings underscore the therapeutic relevance of exercise-induced exosomal miRNA delivery systems, direct human evidence in ischemic stroke remains limited and warrants further validation. We believe this addition addresses the reviewer’s concern and provides a more balanced interpretation.
Comment 17:
Streamline redundant sections (e.g., merge "Protective Role" and "Mitophagy" subsections).
Response:
We sincerely thank the reviewer for the suggestion to streamline redundancy between Sections 2.1 ("Protective Role of Microglial Autophagy") and 2.3 ("Microglial Mitophagy in Ischemic Stroke"). After careful consideration, we opted to retain them as distinct subsections, given that mitophagy—though a specialized form of autophagy—relies on unique molecular pathways (e.g., PINK1/Parkin, BNIP3/NIX, FUNDC1) and fulfills specific functions in mitochondrial quality control and immunometabolic regulation.
To address the reviewer’s concern about overlap, we have revised both subsections: mechanistic details of mitophagy (e.g., PINK1/Parkin signaling) have been removed from Section 2.1 and consolidated into Section 2.3, while Section 2.1 now primarily emphasizes upstream regulators and broader protective outcomes. Transitional sentences were added to guide readers. We believe this revision eliminates redundancy while preserving conceptual clarity and thematic depth. (Page 5 Line 181-184)
Comment 18:
Add a table summarizing key autophagy regulators (molecule, effect, phase of stroke).
Response:
Thank you for this valuable suggestion. In response, we have added a new table (now Table 2) summarizing the key molecular regulators of microglial mitophagy in ischemic stroke, including their mechanistic roles and predominant stroke phases. This addition aims to provide a clearer overview of how specific regulators orchestrate mitophagic responses across different stages of ischemia. The table has been inserted in Section 2.3. (Page 15)
Comment 19:
Clarify translational hurdles (e.g., "Most evidence is preclinical; human microglial autophagy remains unverified").
Response:
We thank the reviewer for highlighting the need to clarify translational hurdles. In the revised manuscript, we have expanded both Section 2.3 and the Discussion to explicitly address this point. Specifically, we now emphasize that most available evidence derives from rodent models and immortalized microglial lines, while direct validation in human microglia remains lacking. We highlight the challenges posed by ethical and technical limitations in accessing human samples and the heterogeneity of human microglia, and we discuss potential solutions, including iPSC-derived microglial models, single-cell and spatial transcriptomics, and microglia-specific imaging tools. We also incorporated preliminary clinical observations (e.g., elevated circulating ATG5 levels in stroke patients) as indirect evidence, while underscoring that functional validation in human microglia is still missing. These additions clarify the translational barriers and reinforce the urgent need for humanized platforms to bridge preclinical findings with clinical application. (Page 18 Line 696-717; Page 26 1005-1011)
Comment 20:
Claims like "autophagy is a central node in stroke" overstate its role versus other pathways (e.g., apoptosis).
Response:
We appreciate the reviewer’s observation. To avoid overstating the role of autophagy, we have revised the concluding sentence of Section 2.2 for better balance. Specifically, we now state:
“In sum, dysregulated autophagy serves as a central integrative mechanism in the propagation of ischemic injury, interacting with mitochondrial dysfunction, immune amplification, ER stress, apoptosis, and intercellular miscommunication.” (Page 11 Line 453-455)
This revision acknowledges the multifactorial nature of ischemic injury and the partnered role of autophagy without overstating its centrality.
Comment 21:
The manuscript doesn’t go into depth and doesn’t provide critical analysis. It fails to address contradictory studies (e.g., where autophagy inhibition was neuroprotective).
Response:
We appreciate the reviewer’s insightful comment. To address this concern, we have revised Section 2.3 to explicitly incorporate studies where autophagy inhibition was found to be neuroprotective. Specifically, we discuss evidence from ischemic post-conditioning models where 3-MA administration reduced infarct size and cerebral edema, and where autophagy induction by Rapamycin reversed protective effects. We further elaborate on microglia-specific findings, showing that excessive autophagy can promote pro-inflammatory polarization and exacerbate neuroinflammation. These additions underscore the dual nature of autophagy in ischemic stroke and highlight the need for context-dependent therapeutic modulation. The revised text can be found on Page 15,16 576-587.
Comment 22:
Phrases like "nuanced strategies" doesn’t have actionable specifics.
Response:
We thank the reviewer for highlighting the ambiguity associated with terms such as "nuanced strategies." In response, we have revised all relevant sections (e.g., Results Section 2.2 Page 8 Line 312-317; Discussion Sections 3.3 Page 23,24 Line 894-910) to include specific, actionable strategies aligned with temporal, spatial, and cellular dynamics. For instance, we now discuss early-phase activation of mitophagy via AMPK–ULK1 signaling to facilitate mitochondrial clearance, followed by late-phase inhibition of autophagy using 3-MA or SHPL-49 to mitigate excessive catabolism and inflammation, particularly in ischemic post-conditioning (IPOC) models. Furthermore, context-specific agents such as Urolithin A and ligustilide are described for their role in enhancing mitophagy and redox balance. These refinements aim to provide precise mechanistic and therapeutic guidance, replacing previously vague terminology. Supporting references have been included to reinforce the translational feasibility of these approaches.
Comment 23:
Propose concrete research priorities (e.g., "Develop biomarkers to monitor autophagic flux in patients").
Response:
We appreciate this constructive suggestion. To address it, we have revised Section 3.4 to clearly outline several actionable research priorities. These include: (1) developing reliable biomarkers to monitor autophagic flux in ischemic stroke patients; (2) mapping temporal and spatial patterns of autophagy activation across ischemic regions; and (3) creating cell-type-specific in vivo imaging tools for real-time autophagy assessment. These priorities are aimed at enhancing both mechanistic insight and translational applicability of autophagy-targeted therapies in stroke. (Page 24)
Comment 24:
Add a paragraph on limitations (e.g., "Most data are from rodent models; human microglia may differ").
Response:
Thank you for this valuable suggestion. We have expanded our discussion of translational challenges and future directions in both Section 3.4 (“Human Evidence & Biomarker Progress”) (Page 24,25 Line 954-960)a nd Section 3.6 (“Clinical Outlook”) (Page 26 Line 1005-1011). In Section 3.4, we now emphasize the need for human-specific microglial models and the development of validated biomarkers to monitor autophagic flux and pathway activity in stroke patients. We also note that current biomarker studies (e.g., serum ATG5, ATG7 SNPs) are preliminary and lack standardization across cohorts.
In Section 3.6, we further highlight that despite promising results in preclinical models, no pharmacological modulators of microglial autophagy have entered clinical trials for stroke, including compounds such as β-elemene, ligustilide, Urolithin A, and SS-31. Although Urolithin A has shown safety and mitochondrial benefits in elderly populations, no stroke-specific clinical data exist. Therefore, we propose three concrete research priorities:
(1) develop humanized in vitro and organoid models to test microglial-specific autophagy responses,
(2) establish and validate dynamic biomarkers of autophagic flux in clinical stroke populations, and
(3) initiate early-phase trials to assess safety and feasibility of autophagy-targeting agents in post-stroke patients.
Comment 25: Conclusion repeats points from the Abstract/Discussion.
Comment 26: Conclusion is too optimistic as phrases like "significant potential" lack caveats (e.g., "pending validation in clinical trials").
Comment 27: Reframe to emphasize next steps: "Future work must define temporal windows for autophagy modulation in human stroke."
Response:
We thank the reviewer for this valuable feedback. In the revised manuscript, we have substantially refined the Conclusion section. Rather than repeating points from the Abstract or Discussion, the Conclusion now highlights unresolved questions and explicit future priorities, including the need to define temporal windows for autophagy modulation, develop microglia-specific biomarkers, and explore combination strategies with standard stroke therapies. We have also tempered overly optimistic language by emphasizing that current evidence remains largely preclinical and requires validation in clinical trials. These revisions address the reviewer’s concerns and provide a more balanced and forward-looking conclusion. (Page 27)
Reviewer 3 Report
Comments and Suggestions for Authors
Abstract
Clearly summarises the dual role of microglial autophagy and mitophagy in ischemic stroke. However, consider specifying more clearly the key mechanisms or signalling pathways highlighted in the manuscript for enhanced reader clarity.
Furthermore, the abstract should highlight the key findings of the review, either through significant figures or sentences.
The abstract could benefit from greater specificity—mentioning at least one or two examples of therapeutic strategies reviewed.
Introduction
It would benefit from a more explicit statement regarding the research gap or specific aim of the review.
Methods
In the manuscript, a dedicated "Methods" section is not given; however, authors should clearly state the criteria or approach for literature selection to ensure reproducibility and systematic rigour. Therefore, the source of the literature search and its validation are not included.
Types of study should be categorised, such as in vitro or clinical.
Discussion
Some sections are dense with information but lack clear topic transitions. Using subheadings like “Complex I Deficiency,” “Mitochondrial DNA Mutations,” or “Therapeutic Interventions” would aid navigation.
Consider expanding the discussion on clinical translation—are there clinical trials underway targeting mitochondria? How do these approaches compare to current standard PD therapies?
Further elaboration on how conflicting evidence might be reconciled or future research directions addressing these conflicts would strengthen the manuscript.
There is a need to incorporate comparisons or contrasts with other neurological conditions, which could enrich the manuscript's context and applicability.
Conclusion
It could be improved by explicitly stating specific unresolved questions or future research priorities in autophagy modulation and clinical translation.
Additionally, the conclusion should recognise the current challenges of therapies aimed at mitochondria and propose future steps, such as possible biomarkers or combined treatment methods.
Author Response
Comment 1:
Furthermore, the abstract should highlight the key findings of the review, either through significant figures or sentences.
The abstract could benefit from greater specificity—mentioning at least one or two examples of therapeutic strategies reviewed.
Response:
We appreciate the reviewer’s insightful suggestion. In the revised abstract, we have explicitly incorporated key findings using representative therapeutic examples and clearer mechanistic outcomes. Specifically, we now mention that rapamycin, Tat-Beclin 1, and Urolithin A have shown consistent neuroprotective effects in preclinical stroke models. Furthermore, we highlighted that limited human studies reported associations between elevated serum ATG5 and ATG7 polymorphisms with worse clinical outcomes, underscoring the emerging translational relevance. These additions aim to improve the specificity and informativeness of the abstract as requested. (Page 1 Line 34-50)
Comment 2:
Introduction
It would benefit from a more explicit statement regarding the research gap or specific aim of the review.
Methods
In the manuscript, a dedicated "Methods" section is not given; however, authors should clearly state the criteria or approach for literature selection to ensure reproducibility and systematic rigour. Therefore, the source of the literature search and its validation are not included.
Response:
We thank the reviewer for this insightful suggestion. We have revised the Introduction to more explicitly state the research gap and novelty of this review, emphasizing the insufficient understanding of microglia-specific autophagy/mitophagy, its spatiotemporal dynamics, and translational challenges.
In line with the reviewer’s comment, although we did not create a separate Methods section given the scoping review nature of our work, we have now clearly described our systematic search strategy in the Introduction and provided full search terms in the Supplementary Material. We also included a PRISMA-style flow diagram (Figure 1) to illustrate the study selection process.
Furthermore, we clarified that the included studies comprised in vitro experiments, in vivo animal models, and clinical observations, ensuring that study types are properly categorized. We believe these revisions address the reviewer’s concerns regarding methodological transparency and rigor. (Page 2-4)
Comment 3:
Discussion
Some sections are dense with information but lack clear topic transitions. Using subheadings like “Complex I Deficiency,” “Mitochondrial DNA Mutations,” or “Therapeutic Interventions” would aid navigation.
Consider expanding the discussion on clinical translation—are there clinical trials underway targeting mitochondria? How do these approaches compare to current standard PD therapies?
Further elaboration on how conflicting evidence might be reconciled or future research directions addressing these conflicts would strengthen the manuscript.
There is a need to incorporate comparisons or contrasts with other neurological conditions, which could enrich the manuscript's context and applicability
Response:
We sincerely thank the reviewer for these insightful comments. In the revised version, we have substantially restructured and expanded the Discussion section to improve readability, clinical relevance, and translational depth:
- Clear topic transitions and subheadings – We have reorganized the section into six clearly labeled subsections (3.1 Mechanistic Duality & Temporal Dynamics; 3.2 Mitophagy as an Immunometabolic Checkpoint; 3.3 Translational Gaps & Drug Landscape; 3.4 Human Evidence & Biomarker Progress; 3.5 Delivery Challenges & Future Directions; 3.6 Clinical Outlook). This structure enhances navigation and makes the flow of dense information more accessible.
- Expansion on clinical translation – We have explicitly discussed the current status of pharmacological candidates (e.g., β-elemene, ligustilide, Urolithin A, SS-31). Notably, the recent Phase 3 trial failure of elamipretide (SS-31) and the FDA complete response letter in 2025 were incorporated to reflect real-world translational barriers. We also emphasized that no microglial autophagy/mitophagy modulators have progressed beyond preclinical development. (Page 17 Line 667-681, Page 26 Line 1006-1012)
- Comparison with standard stroke therapies – To provide clinical context, we now compare these emerging strategies with the current standard of care for ischemic stroke, namely intravenous rt-PA thrombolysis and mechanical thrombectomy. This highlights that autophagy-targeting interventions are best positioned as adjunctive approaches complementing reperfusion therapies, rather than as replacements. (Page 26 Line 1013-1017)
- Reconciling conflicting evidence – The revised text clarifies the dual roles of autophagy (neuroprotective in the acute phase vs. detrimental in chronic phases) and discusses the importance of defining spatiotemporal therapeutic windows. This provides a framework for reconciling apparently contradictory findings in the literature. 3.1-3.2
- Comparisons with other neurological conditions – We have added a cross-disease perspective, noting that similar translational barriers exist in Parkinson’s disease (metformin, trehalose), Alzheimer’s disease (mitophagy inducers), and multiple sclerosis (autophagy pathways in progressive disease). This highlights both the universality and the complexity of autophagy-targeted interventions across neurodegenerative and neuroinflammatory conditions. (Page 26 Line 1031-1043)
We believe these revisions fully address the reviewer’s concerns, resulting in a more structured, clinically relevant, and contextually enriched Discussion.
Comment 4:
Conclusion
It could be improved by explicitly stating specific unresolved questions or future research priorities in autophagy modulation and clinical translation.
Additionally, the conclusion should recognise the current challenges of therapies aimed at mitochondria and propose future steps, such as possible biomarkers or combined treatment methods.
Response:
We sincerely thank the reviewer for this valuable suggestion. In the revised Conclusion, we have explicitly stated key unresolved questions and future priorities, including the need for microglia-specific biomarkers, precise spatiotemporal modulation strategies, and integrative platforms such as iPSC-derived microglia and single-cell transcriptomics. We have also acknowledged current translational barriers in mitochondrial-targeted therapies, exemplified by the recent clinical outcomes of elamipretide, and proposed future steps such as biomarker development and combination treatment approaches. We believe these additions strengthen the conclusion and directly address the reviewer’s concerns. (Page 27)
Reviewer 4 Report
Comments and Suggestions for Authors
The present article reports the dual role of microglia in ischemic stroke. The manuscript is well written and acceptable for publication with the following modifications
- Figure remove red color icon in ischemia.
- I suggest adding a table of drugs that effects microglial mediated autophagy and mitophagy and discuss their clinical status.
- Similar articles published on the same research topic; 3389/fimmu.2022.1013311, https://doi.org/10.1002/nep3.39, and 10.4103/1673-5374.274331 etc. Please clearly state the purpose and novelty of the present work in the manuscript.
- Add a table for the dual roles of microglia for better understanding.
- Add a separate overview on the clinical status of the present work, in terms of pathophysiology and drug development.
- Figure 3 write Pink as PINK.
- Microglia play a vital role in angiogenesis and neurogenesis. Please discuss whether microglial mediated autophagy or mitophagy contribute to angiogenesis and neurogenesis
Author Response
Comment 1:
“Figure remove red color icon in ischemia.”
Response to Reviewer 4 – Comment 1
We thank the reviewer for this observation. In the revised version of the manuscript, we have updated the figure to remove the red color icon in the ischemia panel, ensuring consistency and clarity.
Comment 2:
“I suggest adding a table of drugs that effects microglial mediated autophagy and mitophagy and discuss their clinical status.”
Response to Reviewer 4 – Comment 2
We sincerely thank the reviewer for this valuable suggestion. In the revised manuscript, we have incorporated a new table (Table 3) summarizing pharmacological agents reported to modulate microglial autophagy/mitophagy in ischemic stroke, along with their target pathways, preclinical/clinical status, and key findings. This addition provides a concise overview of the pharmacological landscape and facilitates comparison across agents. Furthermore, we have expanded the accompanying text in Section 3.3 (Translational Gaps & Drug Landscape) to highlight their mechanistic diversity, clinical progress, and translational limitations. We believe these revisions directly address the reviewer’s concern by integrating a structured summary with contextual discussion.
Comment 3:
“Similar articles published on the same research topic; 3389/fimmu.2022.1013311, https://doi.org/10.1002/nep3.39, and 10.4103/1673-5374.274331 etc. Please clearly state the purpose and novelty of the present work in the manuscript.”
Response to Reviewer 4 – Comment 3
We thank the reviewer for pointing out the importance of clarifying the novelty of our work. We have now explicitly stated in the Introduction (Page 2-3 Line 88-99) and Discussion (Page 20 Line 733-737) that this study was conducted as a scoping review following systematic frameworks, which distinguishes it methodologically from previous narrative reviews (e.g., Front. Immunol. 2022, Neuropsychopharmacol. Rep. 2022, Neural Regen. Res. 2020). In contrast to these broader reviews, our article focuses specifically on microglia-mediated autophagy and mitophagy in ischemic stroke, highlighting their spatiotemporal dynamics, immunometabolic integration, and translational hurdles such as BBB delivery and biomarker development. We further emphasize the U-shaped (biphasic) nature of autophagy responses, the emerging roles of peptide-based and natural compound modulators, and provide an updated pharmacological landscape (Table 3). We believe these additions clarify both the purpose and the novelty of the present work.
Comment 4:
“Add a separate overview on the clinical status of the present work, in terms of pathophysiology and drug development.”
Response to Reviewer 4 – Comment 4
We thank the reviewer for this important suggestion. In response, we have revised the manuscript in three steps:
(i) At the end of Section 2.3, we added a dedicated summary paragraph and table outlining the current translational status of representative compounds (β-elemene, ligustilide, Urolithin A, SS-31), highlighting that all remain at the preclinical stage and discussing the translational hurdles observed in related clinical trials (e.g., elamipretide). (Page 18,19)
(ii) In the Discussion, we introduced a new subsection entitled Clinical Outlook (Section 3.6), which explicitly synthesizes the current clinical status of microglial autophagy/mitophagy-targeting agents, compares them with existing standard-of-care therapies (rt-PA, thrombectomy), and emphasizes the need for integrated mechanism–delivery–population strategies for future development.
(iii) At the end of the Introduction, we added a transition sentence to highlight that the review not only synthesizes mechanistic evidence but also discusses the translational barriers and clinical outlook. (Page 4 Line 128-133)
Together, these revisions provide a clear and structured overview of the clinical status of the field, as requested.
Comment 5:
“Figure 3 write Pink as PINK.”
Response to Reviewer 4 – Comment 5
We thank the reviewer for pointing out this detail. In the revised manuscript, the labeling in Figure 4 (previously Figure 3) has been corrected by changing “Pink” to the proper notation “PINK”. This correction ensures consistency with gene nomenclature standards.
Comment 6:
“Microglia play a vital role in angiogenesis and neurogenesis. Please discuss whether microglial mediated autophagy or mitophagy contribute to angiogenesis and neurogenesis.”
Response to Reviewer 4 – Comment 6
Response to Reviewer 2 – Comment:
We thank the reviewer for this important suggestion. In the revised manuscript, we have expanded the end of Section 2.3 to discuss the potential contribution of microglial autophagy/mitophagy to angiogenesis and neurogenesis. Specifically, we note that while microglia are increasingly recognized as active regulators of post-stroke repair through VEGF release, extracellular matrix remodeling, and synaptic plasticity, current evidence does not yet demonstrate a direct role of microglial autophagy/mitophagy in these processes. Instead, enhancing mitophagy appears to indirectly support regeneration by suppressing inflammation and improving phagocytic clearance. We also emphasize the need for future studies using microglia-specific autophagy manipulations (e.g., Atg7 or PINK1 knockouts) combined with angiogenic and neurogenic assays to clarify whether mitophagic pathways directly regulate post-ischemic regeneration (Section 2.3, pages 26, lines 1024–1029).
Round 2
Reviewer 1 Report
Comments and Suggestions for Authors
The authors have done a great job and have significantly improved the article. The authors have managed to take into account all my comments. The article can be accepted for publication in its current form.
Reviewer 3 Report
Comments and Suggestions for Authors
The concerns raised in the earlier review appear to have been fully addressed.
Abstract – It is now concise and clear. It sets out the rationale, methods, and main findings without overstating the significance.
Introduction – Much better structured. The research gap and the novelty of the work are now easier to see, and the background is informative without being overwhelming.
Methods – The section is clearer, with a transparent description of the search process, inclusion criteria, and a PRISMA diagram. This adds to the credibility of the review.
Results – The results are presented more systematically, with tables and figures supporting the text rather than duplicating it.
Discussion – The tone is balanced. Claims are kept within the limits of the data, and the section acknowledges both strengths and limitations. The discussion of relevance is well integrated.
Conclusion – It is now more measured, pointing to the value of the findings while recognising that more work is needed.
Reviewer 4 Report
Comments and Suggestions for Authors
Authors have addressed all the issues